# How the representation of microphysical processes affects tropical condensate in the global storm-resolving model ICON

Ann Kristin Naumann[1,2], Monika Esch[1], and Bjorn Stevens[1]

[1]Max Planck Institute for Meteorology, Hamburg, Germany
[2]Meteorological Institute, Center for Earth System Research and Sustainability (CEN), Universität Hamburg, Hamburg, Germany

**Correspondence:** Ann Kristin Naumann (ann-kristin.naumann@mpimet.mpg.de)

**Abstract.** Cloud microphysics are a prime example of processes that remain unresolved in atmospheric modelling with storm-resolving resolution. In this study, we explore how uncertainties in the representation of microphysical processes affect the tropical condensate distribution in a global storm-resolving model. We use ICON in its global storm-resolving configuration, with a one- or a two-moment microphysical scheme and do several sensitivity runs, where in each we modify parameters of one hydrometeor category of the applied microphysics scheme. Differences between the one- and the two-moment scheme are most prominent in the partitioning of frozen condensate in cloud ice and snow, and can be understood in terms of the habit's definition for each scheme which is associated with different process rates. Overall differences between the simulations are moderate and tend to be larger for individual condensate habits than for more integrated quantities, like cloud fraction or total condensate burden. Yet, the resulting spread in the tropical energy balance of several W m$^{-2}$ at the top of the atmosphere and at the surface is substantial. Although the modified parameters within one scheme generally affect different process rates, most of the change in the condensate amount of the modified habit and even total condensate burden can be attributed to a single property, the change in fall speed. Tropical mean precipitation efficiency is also well explained by changes in the relative fall speed across different habits and both schemes.

## 1 Introduction

As global storm-resolving models (SRMs) begin to resolve several processes that climate models previously had to parameterize (Satoh et al., 2019; Stevens et al., 2019), those processes that must still be parameterized, e.g., cloud microphysics and subgrid-scale turbulence, become a more important source of uncertainty (Lang et al., 2023). In the case of cloud microphysical processes, whereby liquid and solid hydrometeor particles span scales ranging from micrometers up to centimeters, this uncertainty is unlikely to disappear even if the trend for refining resolution continues. Nonetheless, because global SRMs begin to resolve the motion fields governing cloud formation, and hence the main factors influencing the cloud microphysical evolution, they differ fundamentally from conventional climate models, whose parametric representations of cloud microphysics must be coupled to parametric representations of clouds and convection. This means that past studies that apply more coarsely resolved climate models (e.g., Sanderson et al., 2008; Yan et al., 2015; Proske et al., 2022) or idealized setups (e.g., McCumber et al., 1991; Hu and Igel, 2023) to explore their sensitivity to the representation of cloud microphysical models may not necessarily

be informative of how the new generation of models will respond to microphysical changes. This motivates the present study, which presents an exploration of microphysical sensitivities in the first global storm-resolving version of ICON (ICOsahedral Nonhydrostatic model; Hohenegger et al., 2023) and investigates the role of microphyscial processes for the tropical condensate distribution therein.

Using output from the DYAMOND (DYnamics of the Atmospheric general circulation Modeled On Non-hydrostatic Do-
mains; Stevens et al., 2019) comparison of global SRMs, Lang et al. (2021) recently show that despite considerable diversity in the representation of cloud microphysical processes, the multi-model spread of tropospheric humidity in the DYAMOND ensemble is reduced compared to conventional climate models. However, cloud fraction and condensate amounts still differ substantially across DYAMOND members (Roh et al., 2021) and accordingly the multi-model spread in outgoing longwave radiation is largest in cloudy regions of the tropics (Lang et al., 2021). Continuous efforts of model development by the NICAM
(Nonhydrostatic ICosahedral Atmospheric Model) group also demonstrate that the microphysics scheme has a strong impact on cloud amount and the radiant energy budget (e.g., Seiki et al., 2014, 2015; Roh et al., 2017). Similarly to conventional climate models, in which cloud ice related microphyscial parameters like the ice fall speed can have a strong effect on climate sensitivity (Sanderson et al., 2008), recent studies demonstrate that ice microphysical processes also play a major role in SRMs, e.g., in controlling the radiant energy budget in the Asian monsoon region (Sullivan and Voigt, 2021; Sullivan et al., 2022) and
the overall tropics (Atlas et al., 2024, using a nudged global SRM).

Uncertainties in the microphysical parameterizations used in global SRMs can arise due to the basic approaches employed, or from uncertainty in how to represent specific processes within a specific approach, or due to limited understanding of the process itself. Common to all approaches in global SRMs is the adoption of bulk parameterizations, which predict a moment, or moments, of an assumed particle-size distribution. One-moment bulk schemes, which typically predict the specific mass, $q_x$, of
the different hydrometeor categories $x$, are commonly used in global SRMs. The NICAM group has also explored the behavior of two-moment schemes (Seiki et al., 2015), whereby the second moment is used to describe the number concentrations of condensate particles, $n_x$. Two-moment schemes allow for more flexibility in the choice of distributions that the hydrometeors are assumed to follow and a process-based variation of condensate number, but come at the cost of greater complexity, a larger number of uncertain parameters, and entail a larger computational burden. Moreover, even two-moment schemes do
not allow sufficient flexibility to describe the full evolution of the hydrometeor distribution under the simplest of processes, such as precipitation (e.g. size sorting; Wacker and Seifert, 2001) or mixing. In fact there is no known parametric distribution whose form remains invariant (and hence whose parameters can be determined by its moments) under known microphysical processes. While a number of studies have demonstrated the advantage of multi-moment schemes over one-moment schemes often in regionally constrained case studies and related to the representation of specific processes (e.g. Sullivan et al., 2023;
Seiki et al., 2015, for a global domain), the situation is less clear for global or more aggregated statistics relevant for climate studies and some efforts have been made to identify processes, or process groups, that dominate the system across schemes (Wacker, 1995; Koren and Feingold, 2011; Mülmenstädt and Feingold, 2018; Proske et al., 2022).

Understanding how to tune the microphysical processes in a global SRM is a daunting task. Given the computational intensity of the simulations, and the shear number of parameters, it is not possible to contemplate a systematic sweep of parameter space,

and given the preliminary state of the model it seems premature to embark on a fine tuning, with systematic comparisons to observations. Here, following the programme outlined by Mauritsen et al. (2022), we began by choosing a few parameters as a first exploration of how one would go about the tuning of a global SRM. We found the simulations unveiled a simple underlying principle: changing habit properties influence how fast a condensate habit falls, or how it transforms into other habits that then fall, thereby determining condensate amount in the atmosphere. In the present study, we hence investigate how different representations of microphysical processes affect the distribution of condensate in the tropical atmosphere in the global SRM ICON, and the implications this has for the distribution of rainfall, and for the radiant energy budget within the tropics.

We implement a two-moment scheme into ICON and consider both differences between it and the default one-moment scheme, and the sensitivity of each of these schemes to the uncertain representation of critical parameters. The two specific schemes applied in this study do not allow for general conclusions about the advantages of one type of scheme over the other, but they are rather interpreted as equally valid but different attempts to represent microphyscial processes in a global SRM. Our findings are based on the analysis of several sensitivity experiments with the atmospheric component of ICON in a global storm-resolving configuration, run with prescribed sea-surface temperatures and sea-ice concentrations. ICON, the simulation setup, the modification for sensitivity runs, and the implications for the fall speeds are presented in Sect. 2. Section 3 analyses differences between the one- and the two-moment scheme and shows that changes in the fall speed alone are largely able to explain differences in condensate load between the simulations. Section 4 focuses on the impact on the energy balance and Sect. 5 on precipitation properties. Finally, Sect. 6 closes with a summary and conclusion.

## 2 Simulations

### 2.1 ICON setup

We conduct several sensitivity experiments with the atmospheric component of ICON (Hohenegger et al., 2023) which solves the Navier-Stokes equation on an icosahedral-triangular C grid. Intended to be run at kilometer-scale grid spacing the atmospheric component of ICON applies parameterizations for three processes: radiation, turbulence, and microphysics. We use *psrad* radiation (Pincus and Stevens, 2013) and Smagorinsky's turbulence scheme (Smagorinsky, 1963) with modifications by Lilly (1962) as implemented by Lee et al. (2022) following Dipankar et al. (2015). Microphysical processes are represented by a one- or a two-moment microphysics scheme as specified in Sect. 2.2.

No parameterization for subgrid-scale clouds is used, leading to a binary cloud fraction of 0 or 1 at each grid point. If the sum of cloud water and cloud ice in an individual grid box is larger than 1 mg kg$^{-1}$ the cloud fraction is set to 1, else to 0. The other hydrometeor categories rain, snow, graupel, and hail do not contribute to cloud fraction. The same demarcation is used in the radiation scheme: only cloud water and cloud ice interact with radiation. By virtue of their assumed smaller size, cloud water and cloud ice interact more strongly with radiation on a per mass basis than would other categories. The neglect of the effect of the other species on the radiant energy budget becomes less well justified as their mass concentrations increase,

particularly if they dominate over the mass concentration of the radiatively active species. For this study this is in particular critical for snow amounts in the one-moment scheme and is discussed in Sect. 4.

All runs apply a quasi-uniform horizontal grid spacing of 5 km and prescribed sea-surface temperatures and sea-ice concentrations. We closely follow the experimental protocol of the DYAMOND intercomparison (Stevens et al., 2019) with initial conditions from IFS (Integrated Forecasting System) analysis at 20th January 2020. An initial simulation with the one-moment scheme is run for 12 days to allow for enough time for model spinup. All analysed sensitivity experiments are restarted at 1st of February 2020 from this simulation and are run for another 10 days. To allow for ample time for spinup, the analysis shown in this study is restricted to the last 5 days of these 10-day simulations and the tropics (30° N to 30° S) if not explicitly stated otherwise. This is a short period for those used to studying climatological effects, but it is long compared to the many case studies used to study the impact of microphysics on regional storm-resolving, or finer scale, simulations (e.g., Xue et al. (2017) for a squall line case, or VanZanten et al (2011) for shallow cumulus). Given the enormous spatial sampling of a global simulation it thus perhaps comes as no surprise that the day-to-day variability of global (or global tropical) statistics is typically smaller than the differences between the simulations (see Fig. 5 and Fig. D1 for the tropical and global top-of-the-atmosphere and surface energy balance). We hence conclude that the simulation length is sufficient to identify systematic effects of changing the representation of microphysical processes.

## 2.2 Microphysical ensemble

We apply two versions of microphysics parameterization as they are also employed in the numerical weather prediction configuration of ICON: a one-moment scheme predicting the specific mass of five hydrometeor categories (cloud water, rain, cloud ice, snow, graupel; Baldauf et al., 2011; Doms et al., 2015) and a two-moment scheme predicting both the specific number and specific mass of six hydrometeor categories (cloud water, rain, cloud ice, snow, graupel, hail; Seifert and Beheng, 2006). While these references give a better idea of the schemes, their implementation is complex and layered by practice. Over 200 parameters alone are used to characterize the one-moment scheme. For this reasons, and because the arguments we develop don't depend on the specific implementation, readers interested in these details are best served by accessing the code itself, which is part of the open source release of ICON. For our purposes it suffices to note that in both schemes, a constant concentration of cloud condensation nuclei is prescribed, and that simulations with the two-moment scheme lead to a total increase in computational time of about 30%, most of which arises from the transport of an increased number of prognostic variables.

For both schemes, the condensate habits are fundamentally defined by their assumed particle-size distribution, the particle-based fall-speed relation, and the rules which specify how they interact with one another and radiant energy transfer. Given the computational intensity of the simulations, and the shear number of parameters, it was not possible to contemplate a systematic sweep of parameter space. Even for a much less computationally intensive model a full sweep of the parameter space would be arduous, and given the amount of structural uncertainty in the models, misleading. For the sensitivity experiments we therefore decided to concentrate on a few parameters arising in a functional characterization of the habits, rather than on the more numerous parameters that tune individual process rates. These properties that define the habits in the bulk schemes affect several of the microphysical process rates at the same time, which can have amplifying or compensating effects. Specifically,

**Table 1.** Overview of simulations described in detail in Appendix A.

| name | description |
| --- | --- |
| 1mom | one-moment scheme with standard parameters |
| 2mom | two-moment scheme with standard parameters |
| 1mom-rain | like 1mom but with a narrower raindrop size distribution |
| 1mom-ice | like 1mom but with a higher ice fall speed |
| 1mom-snow | like 1mom but with lower-density snow |
| 2mom-rain | like 2mom but with a modified $\mu$-$D$ relationship |
| 2mom-ice | like 2mom but with a higher ice fall speed |
| 2mom-snow | like 2mom but with modified snow properties |

we apply one of the two microphysics schemes and modify parameters of one hydrometeor category in that scheme over what we deem to be a plausible range. We perform eight simulations in total and the sensitivity experiments are named according to the microphysics scheme and the hydrometeor category that is modified, e.g., 1mom-ice. An overview of the simulations is given in Tab. 1 and the specific modifications are detailed in Appendix A. The choice of parameters to perturb and the

130 magnitude of their perturbation reflects our judgement of parameters whose plausible range of values is likely to have the largest impact on the simulations and is guided by tuning choices of earlier versions of the model and literature values. Given that the standard parameters of the one- and two-moment scheme are inherited from other simulation setups and have not been tuned for global kilometer-scale simulations, we interpret all eight simulations as equally plausible realisations, with no preference of one over the other.

By modifying parameters of the functional characterization of the habits, the effect is not limited to a single process rate but generally affects many different process rates of that condensate's habit. However, all modifications directly or indirectly affect the fall speed of the specific mass of the modified condensate type, and in the case of the two-moment scheme the fall speed of the specific number of the modified condensate type (Fig. 1). In turn, a habit's fall speed of the specific mass (and number) affects the sedimentation flux of that habit but also other process rates, like riming or aggregation. An exception is 1mom-ice

where we modify the fall speed of the specific ice mass (instead of the particle-based fall speed) which, in the formulation of the particular one-moment scheme we use, affects solely the sedimentation flux of ice and none of the conversion rates. For the remainder of the manuscript, we will use *fall speed* as synonym for *fall speed of the specific mass* if not explicitly stated otherwise. We will argue later that the change in condensate amount is largely determined by the change in fall speed, even for those modifications that affect the fall speed indirectly.

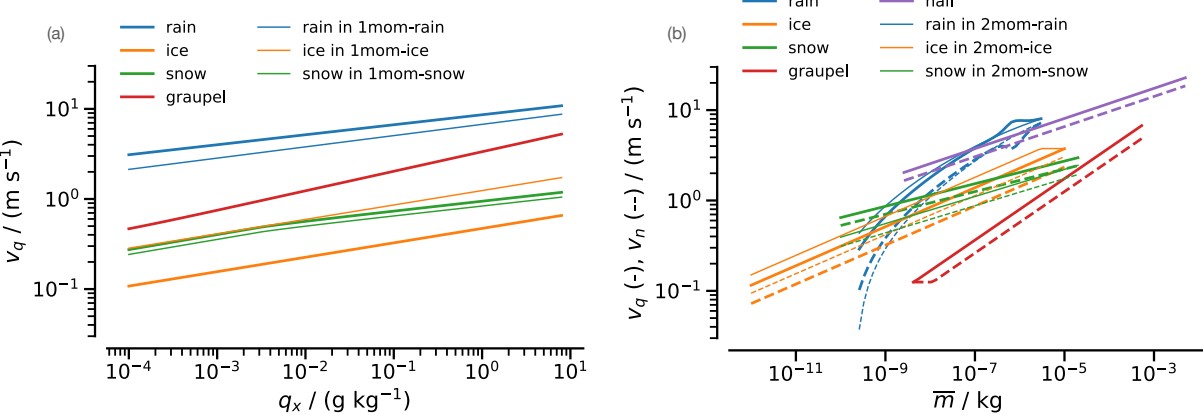

**Figure 1.** How modifications in the sensitivity simulations affect the fall speeds. (a) In the one-moment scheme the fall speed of the specific mass, $v_q$, is a function of the specific mass, $q_x$. (b) In the two-moment scheme the fall speed of the specific mass, $v_q$ (solid), and the fall speed of the specific number, $v_n$ (dashed), are a function of mean particle mass, $\overline{m} = q_x/n_x$. All fall speeds depend on air density, $\rho$, and the fall speed of snow in the one-moment scheme has a slight dependence on temperature, $T$. Fall speeds shown here are calculated for $\rho = 0.7$ kg m$^{-2}$ and $T = 253$ K. Fall speeds of different habits are shown in different colours as indicated by the legends. Thick lines represent the default values of the schemes used in this study for the control runs (i.e., 1mom, and 2mom), while thin lines represent the modified fall speed of a particular habit in a particular sensitivity run (e.g., thin blue line for the modified rain fall speed in 1mom-rain and 2mom-rain.)

## 145  3   Sensitivity of condensate to the representation of microphysics

The first thing we learn from the simulations is that the changes do not lead to drastic differences in condensate amount (Fig. 2). All show a trimodal cloud fraction structure, i.e., three maxima in cloud fraction (shallow, mid-level, and deep convection), albeit more pronouncedly so for the two-moment scheme, and important quantities such as the profile of relative humidity appear insensitive to the microphysical representation. Differences between the one-moment and two-moment scheme tend to be larger than differences resulting from parameter perturbations in a single scheme, and changes in individual condensate habits tend to be larger than for more integrated quantities, like cloud fraction or total condensate burden. Not unsurprisingly, changes to the representation of one habit in a particular scheme also tend to show up most in the representation of that habit. For example, in both the two- and one-moment scheme increasing the ice fall speed reduces the amount of cloud ice, but a similarly direct influence of a change to a habit, and the burden of that habit, is also evident for the other habits. A greater sensitivity to the type of (one- versus two-moment) scheme suggests that the parameter changes were conservatively chosen, and thus unlikely span the full range of uncertainty.

### 3.1   Differences between schemes

From the point of view of the condensate burdens, the largest changes are evident in the partitioning of frozen condensate between ice and snow in the upper troposphere. The one-moment scheme produces much more snow, and much less ice,

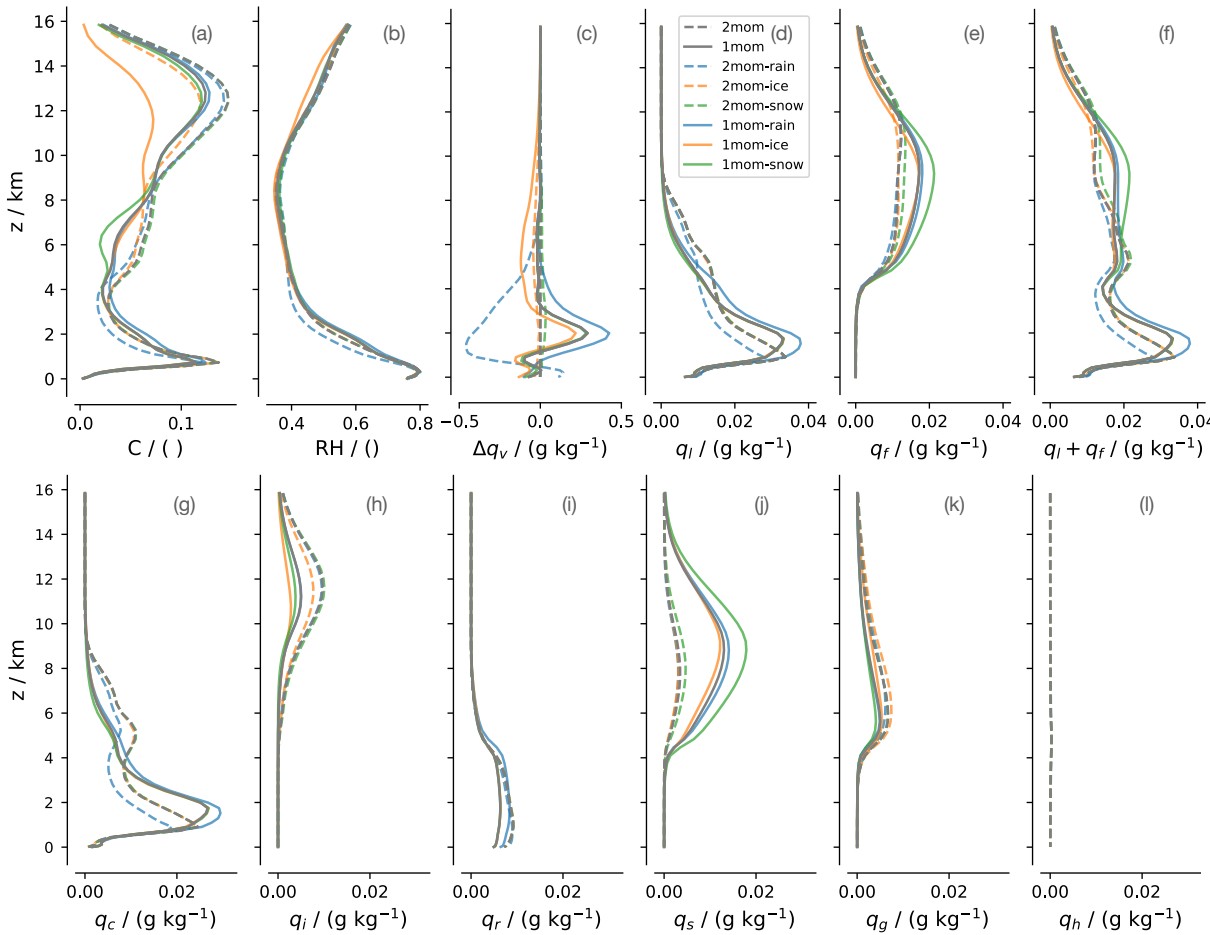

**Figure 2.** Tropical mean profiles of (a) cloud fraction, $C$; (b) relative humidity, RH; (c) specific water vapor mass as difference to 2mom; specific mass of (d) liquid condensate, $q_l = q_c + q_r$; (e) frozen condensate, $q_f = q_i + q_s + q_g + q_h$; (f) total condensate, $q_l + q_f$; and for each hydrometeor category: (g) cloud water, $q_c$; (h) cloud ice, $q_i$; (i) rain, $q_r$; (j) snow, $q_s$; (k) graupel, $q_g$; and (l) hail, $q_h$.

than the two-moment scheme (Fig. 2). In the one-moment scheme snow is, in fact, the dominant frozen hydrometeor habit throughout the upper troposphere (above 6 km). In the two-moment scheme the habit burdens are more equally partitioned, albeit distributed by height, or temperature, and the two-moment scheme allows for more supercooled cloud water above 6 km.

The one-moment scheme produces high condensate burdens of snow more often while ice typically occurs at lower burdens than in the two-moment scheme (Fig. 3). This hints at processes that remove high ice burdens more effectively but allow for high snow burdens in the one-moment scheme. Indeed this is consistent with the one-moment scheme in this study interpreting ice and snow as separated by size. Therefore, ice in the one-moment scheme cannot become large and deposition-autoconversion will turn high ice concentrations into snow. In the two-moment scheme in this study ice and snow are differentiated by morphology (monomers versus aggregates) and hence deposition-autoconversion is not considered. In addition there are more

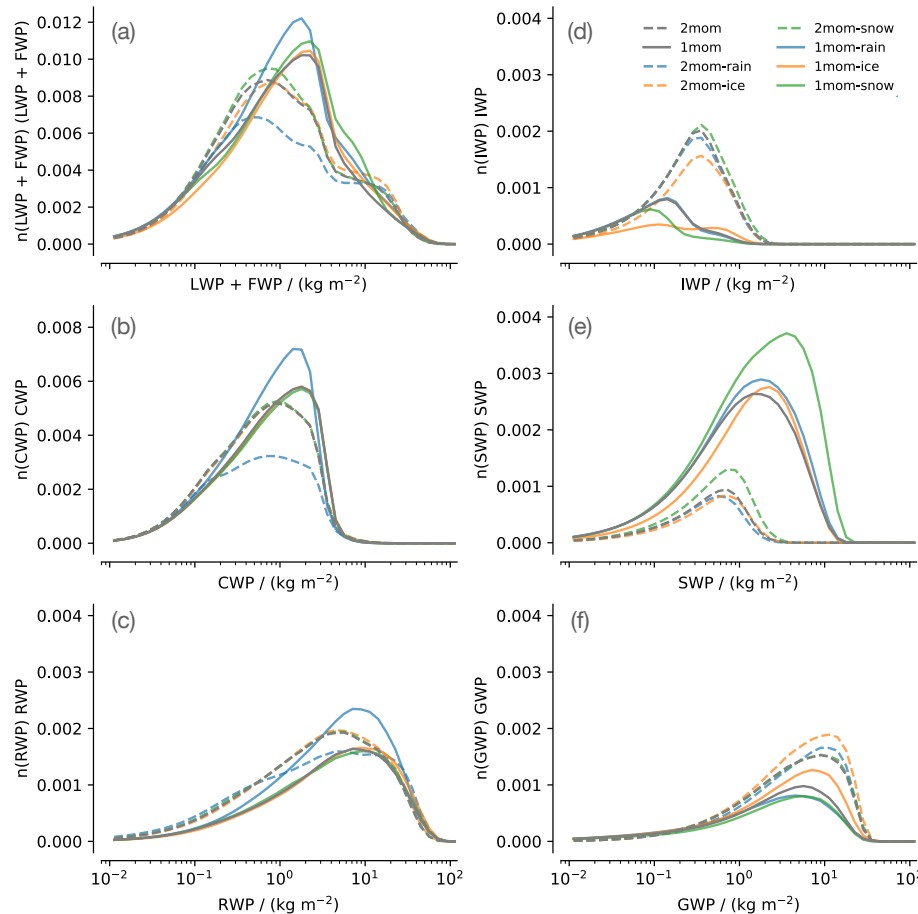

**Figure 3.** Distribution of vertically integrated condensates in the tropics: (a) total condensate path as the sum of liquid and frozen water path, LWP + FWP; (b) cloud water path, CWP; (c) rain water path, RWP; (d) ice water path, IWP; (e) snow water path, SWP; and (f) graupel water path, GWP. For the distributions the specific number of columns in each bin, $n$, is multiplied by the bin value, so that the area under the curve is proportional to the amount of water. Due to its low overall amount hail is omitted in this figure (see Fig. 2 l). While (c-f) share the same $y$-axis, please note the different scales for (a) and (b).

processes that act as sinks for snow in the two-moment scheme keeping snow burdens limited: riming and ice multiplication can turn snow into ice again, and there are additional variants of melting, aggregation, and riming to remove snow. The latter two are sources for graupel which might explain the higher abundance of graupel in the two-moment scheme.

These differences in formulating process rates are specific to the two schemes used in this study and do not reflect intrinsic differences between any one- and two-moment schemes. Our results, however, agree with earlier studies on limited-domains or in idealized setups on the more general result, that the specific definitions of habits and their properties affect the development of convective systems (e.g., McCumber et al., 1991; Adams-Selin et al., 2023).

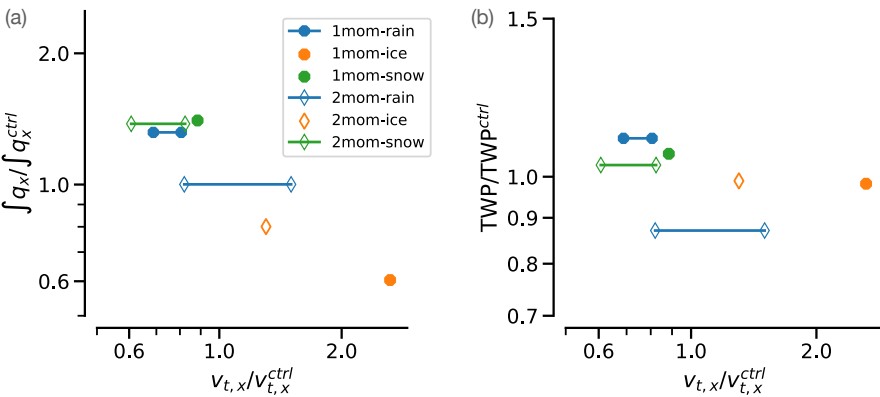

**Figure 4.** (a) Normalized change in tropical mean vertically integrated habit amount, $\int q_x / \int q_x^{\mathrm{ctrl}}$, as a function of the normalized change of that habit's fall speed, $v_{t,x}/v_{t,x}^{\mathrm{ctrl}}$. (b) Normalized change in tropical mean vertically integrated total condensate amount, TWP / TWP$^{\mathrm{ctrl}}$, as a function of normalized change of the modified fall speed. Normalization is done separately for simulations applying the one- and the two-moment scheme with values from 1mom and 2mom, respectively. For 1mom-rain, 2mom-rain and 2mom-snow the change in fall speed varies with $q_x$ or $\overline{m}$ (Fig. 1) and is given as a range. All variables are plotted on a logarithmic axis so that e.g. a doubling and a halving of the fall speed or the condensate amount shows up as the same size of an effect.

### 3.2 Fall speed modifications dominate change in condensate burden for each scheme

Changes to the representation of one habit in a particular scheme generally affect different process rates of that condensate's habit but we find that the implications of the modifications on a single property, the fall speed (see Fig. 1), largely determine the change in the condensate amount of the modified habit: the faster a condensate falls, the less there is of it (Fig. 4). This change is consistent for both schemes. For example, a moderate increase in ice fall speed in the two-moment scheme leads to a moderate decrease in cloud ice, while a stronger increase in ice fall speed in the one-moment scheme leads to a proportionally stronger decrease in ice burden. Roughly, this behavior even holds across condensate habits.

This relation between a change in fall speed and the change in condensate burden also holds regionally with 79 to 96 % of the tropical or global area seeing an increase of condensate habit with decreasing fall speed and vice versa (Appendix B). The absence of strong geographic variations in the response to microphysical perturbations gives us confidence that global responses (as discussed in Fig. 2 and Fig. 4) are informative. On the other hand it also indicates that relatively little regional space needs to be covered in small domain case studies to anticipate global responses. The regional consistency is not seen in 2mom-rain, where the modification of the shape of the raindrop size distribution leads to a wide range of change of rainwater fall speeds including both an increase and a decrease depending on mean raindrop size. The results for this simulation are therefore inconclusive for the tropical mean, which is confirmed by regional ambiguity.

In addition, we find that the change in fall speed also affects the total amount of condensate, although to a lesser degree than the change in condensate amount of the modified habit (Fig. 4). This is because the condensate burden of the modified habit is only a part of the total condensate amount but also indirect changes of condensate amount of the other habits tend to

compensate for the change in the modified habit. For example, for a reduced snow fall speed in the one-moment scheme, ice is reduced (Fig. 2, Fig. 3), probably because accretion of ice by snow becomes more effective, when snow amount increases. A narrower raindrop size distribution in the one-moment scheme reduces sedimentation of rain water and therefore increases the rainwater amount. This increases evaporation from rain water and hence specific humidity, which allows for more cloud water.

2mom-rain again behaves somewhat differently in that the largest changes are not in the modified rain habit but in a decrease of the cloud water burden. For 2mom-rain large raindrops that fall out effectively experience weaker size sorting which implies that rainwater remains suspended in air for longer but is distributed over fewer raindrops, hence, increasing mean raindrop size and counteracting low fall speeds. Indeed the mean rainwater profile is close to 2mom but the increase in mean raindrop size reduces the efficiency of evaporation leading to a drier and warmer lower troposphere which inhibits easy cloud formation and hence reduces cloud water (Fig. 2). Note that while the rain water response in 2mom-rain is not visible in the mean profiles, it does affect the distribution of rainwater burdens (Fig. 3) and the modified evaporation plays a role for the shift in surface precipitation and precipitation efficiency discussed in Sect. 5.

## 4  Energy balance

While condensate differences between simulations with a one-moment scheme and a two-moment scheme are generally larger than among the simulations with perturbed parameters of one specific scheme, this is not the case for the energy balance (Fig. 5). Given that the modifications of the sensitivity runs are moderate, the overall spread of several W m$^{-2}$ at the top of the atmosphere (TOA) and at the surface are substantial. As for condensate amount, the day-to-day variability is smaller than differences between the simulations which indicates that the signals are robust, and short simulations are informative, which is in particular relevant for future tuning endeavours.

The standard deviation of the net tropical TOA energy balance in our ensemble is 2.6 W m$^{-2}$. For all simulations differences in the individual components, shortwave upward, SW↑, and longwave upward flux, LW↑, are larger but partly compensate, more strongly so if high-cloud effects dominate over low-cloud effects. Across schemes, perturbations of individual habits have consistent effects on the energy budget: Less cloud ice (as in 1mom-ice and 2mom-ice) leads to less SW↑ and more LW↑ with a small net cooling of the atmospheric column. More cloud water (as in 1mom-rain) leads to more SW↑ and slightly less LW↑ resulting in a net cooling of the atmospheric column; and vice versa for less cloud water in 2mom-rain.

Differences in the radiative balance resulting from parameter perturbations in a single scheme may be due to differences in cloud fraction or to changes in radiative properties of cloudy or clear-sky points. We find that profiles of cloud fraction differ little between the schemes, but the condensate habits considered to contribute to cloud fraction, cloud water and cloud ice, differ substantially (Fig. 2). In the two-moment scheme mean cloud-ice loads are substantially larger but also occur more often in higher concentrations so that high-cloud fraction is just a little larger than in the one-moment scheme (Fig. 3). Vice versa, mean cloud water is larger for the one-moment scheme but in combination with higher abundance of high cloud-water concentrations results in virtually the same low-cloud fraction as for the two-moment scheme. Cloud fraction is also more robust to parameter changes within one scheme than the changes in cloud water and cloud ice indicate. A formal decomposition of the components

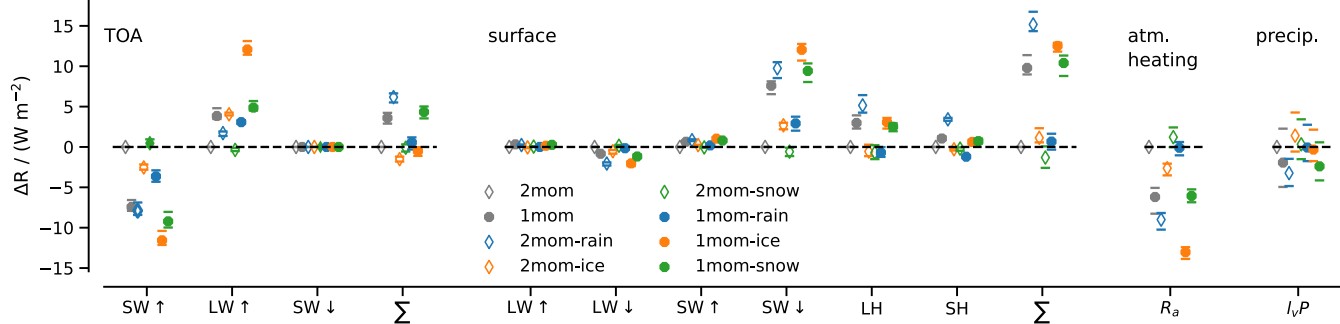

**Figure 5.** Contributions to the tropical (a) top-of-the-atmosphere (TOA) and (b) surface energy balance as well as (c) atmospheric heating and surface precipitation as difference to 2mom. At TOA $\sum$ = SW↓ - SW ↑ - LW ↑> 0 W m$^{-2}$ is a net energy transport into the atmosphere at TOA. At the surface $\sum$ = LW ↓ - LW ↑ + SW ↓ - SW ↑ + LH + SH> 0 W m$^{-2}$ is a net energy transport into the surface. Atmospheric heating is $R_a = \sum_{\text{TOA}} - \sum_{\text{surface}}$. Markers show the 5-day mean and error bars indicate the range from the minimum to the maximum daily mean within the 5-day period. SW: shortwave flux, LW: longwave flux, ↑: upward, ↓: downward, LH: latent heat flux, SH: sensible heat flux, $l_v P$: latent heat of vaporization times surface precipitation.

contributing to the TOA energy balance confirms that the small changes in cloud fraction play a minor role and that the change in radiative properties of cloudy points which is related to the shift in condensate concentrations is dominating (see Appendix C). An exception is the simulation with increasing ice fall speed in the one-moment scheme: Less cloud-ice amount and a shift

towards higher cloud-ice concentrations lead to a strong reduction in high-cloud fraction, which dominates the change in the radiative balance.

In the applied radiation scheme only cloud water and cloud ice interact with radiation, while the radiative effects of rain, snow, graupel, and hail are ignored. This makes the generalizations of the radiative imprint beyond their interpretation for ICON harder and might play a role in particular in the simulations with the one-moment scheme, which have more of their frozen

condensate in the form of snow. Taking into account the radiative effect of snow would tend to increase SW↑ and decrease LW↑ at TOA, we expect the difference in SW↑ and LW↑ to be somewhat overestimated for simulations with the one-moment scheme compared to a simulation with the two-moment scheme but the effect on the net remains unclear although it is probably small due to the tendency of SW↑ and LW↑ to compensate each other. Within the one-moment ensemble, modifications of 1mom-snow have the strongest effect on snow amount. Compared to 1mom, the radiative effect of less ice in 1mom-snow would tend

to be counteracted by the radiative effect of more snow, i.e., differences between the two simulations might be overestimated.

The standard deviation of the net tropical surface energy balance in our ensemble is 6.1 W m$^{-2}$ which is more than twice as large as at TOA. The spread is dominated by the spread in downward shortwave flux, SW↓, (standard deviation $\sigma = 4.5$ W m$^{-2}$) with some modifications from latent and sensible heat flux. Differences in SW↓ at the surface are closely related to differences in SW↑ at TOA, which the inclusion of snow would tend to reduce. They also set differences in the atmospheric

heating rates ($\sigma = 4.7$ W m$^{-2}$) which are known to affect the circulation and spatial distribution of precipitation (e.g., Slingo

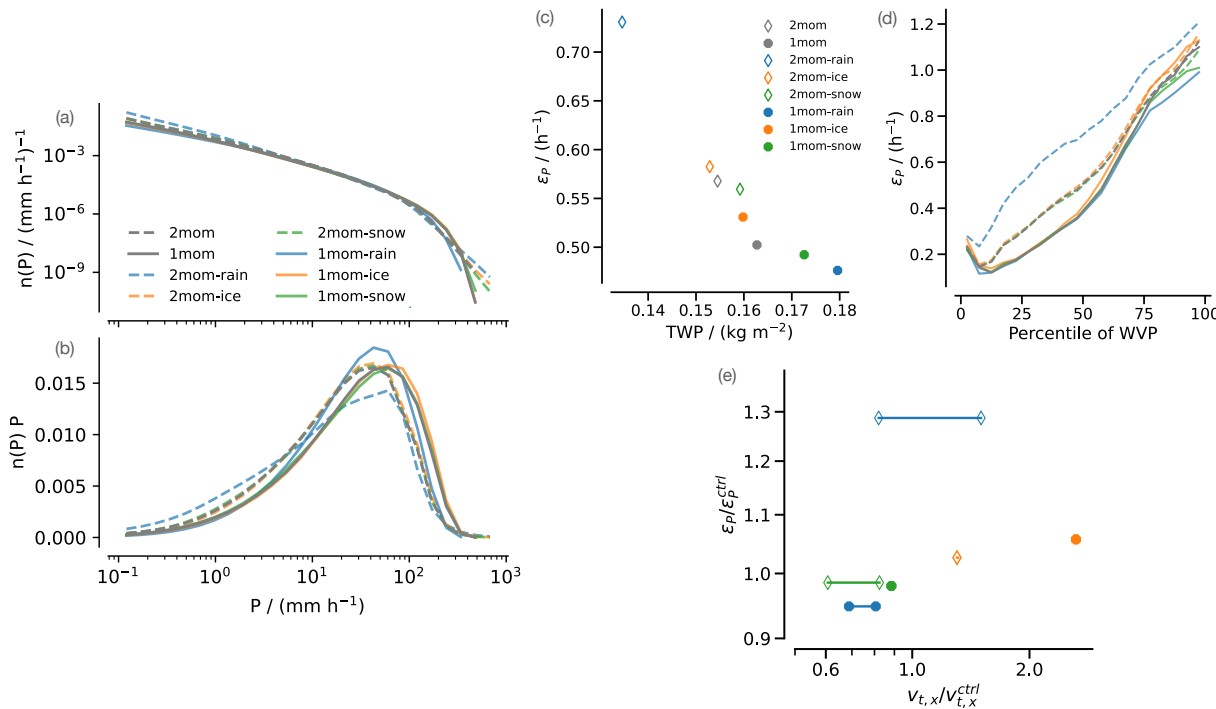

**Figure 6.** Characteristics of tropical surface precipitation, $P$: (a) probability density function $n(P)$; (b) as (a) but multiplied by the bin value so that the area under the curve is proportional to the total amount of $P$; (c) relation of the tropical mean total water path, TWP, and precipitation efficiency, $\varepsilon_P$; (d) $\varepsilon_P$ as a function of percentiles of water vapor path, WVP; (e) normalized $\varepsilon_P$ as a function of relative modification of fall speed.

and Slingo, 1988). Qualitatively similar conclusions can be drawn for the global energy balance but differences among the simulations tend to be a bit smaller than in the tropics (net global TOA energy balance: $\sigma = 2.5$ W m$^{-2}$; Appendix D).

## 5  Precipitation properties

Tropical mean surface precipitation does not robustly differ among the runs (Fig. 5). For all runs except 2mom-rain, the effect
of microphysical model choices is smaller than the day-to-day variability in a five day period. Radiative-convective equilibrium prescribes that the longterm global average precipitation is restricted by the radiative cooling rates, which limits how much precipitation can vary on average on long time scales.

Locally and over shorter periods this restriction does not apply. Grid-scale, instantaneous, tropical precipitation, $P$, below 50 mm h$^{-1}$ is more frequent in 2mom than in 1mom while strong precipitation above 50 mm h$^{-1}$ is more frequent in 1mom
(Fig. 6 a, b). Difference in precipitation properties are larger between the one-moment ensemble and the two-moment ensemble than among each ensemble, and generally small for the latter. Largest differences are found when a parameter of the raindrop size distribution is modified (1mom-rain, 2mom-rain) and changes in the distribution of $P$ are roughly in line with the change

in the distribution of the rain water path (Fig. 3). For 2mom-rain, we also find an increase in frequency of low $P$, which is consistent with a decrease in evaporation affecting light precipitation more strongly. Modifying the frozen hydrometeor

parameters in 1mom/2mom-ice and 1mom/2mom-snow does not substantially affect the distribution of precipitation.

The representation of microphysical processes affects how efficient precipitation is able to form in clouds. We define precipitation efficiency as the fractional amount of total condensate in a $1° \times 1°$ column with TWP $> 10^{-3}$ kg m$^{-2}$ that is returned to the surface as precipitation per unit time, $\varepsilon_P = P/\text{TWP}$, where the total water path, TWP, is the vertically integrated sum of all condensates (e.g., Lau and Wu, 2003; Radtke et al., 2023). The inverse of $\varepsilon_P$ therefore describes a typical residence time

of condensate in the atmosphere. Tropical mean $\varepsilon_P$ varies between 0.48 h$^{-1}$ (1mom-rain) and 0.73 h$^{-1}$ (2mom-rain) among the simulations (Fig. 6 c). Tropical mean TWP decreases with increasing $\varepsilon_P$ and the relation exhibits a very strong correlation (Pearson correlation coefficient: -0.95). The amount of atmospheric condensate is hence largely set by $\varepsilon_P$.

The two-moment runs generally have a higher mean $\varepsilon_P$ and lower TWP than the one-moment runs. Among the two-moment runs, 2mom-rain stands out with a particular large $\varepsilon_P$ and low TWP. Per definition $\varepsilon_P$ increases with decreasing TWP for

constant $P$. For 2mom-rain this increase in mean $\varepsilon_P$ is even stronger than the decrease in *mean* TWP at constant *mean* $P$ would imply (not shown) which points to a correlation between local TWP and $P$. Indeed, $\varepsilon_P$ increases in all regions but this increase is particularly strong in the dry, subtropical regimes (Fig. 6 d, low WVP), which is consistent with a more frequent occurrence of low $P$ (Fig. 6 a, b) and less evaporation in 2mom-rain (see discussion in Section 3.2). Low-intensity precipitation rates play a minor role for the overall precipitation mass which is limited by radiation but they dominate the precipitation area

which affects mean precipitation efficiency.

For the same reasons, 2mom-rain also stands out when the change in fall speed is related to the change in $\varepsilon_P$ (Fig. 6 e). Among the other simulations we generally find that the faster a condensate falls, the higher the precipitation efficiency but there relative changes in $\varepsilon_P$ are well below 10 %.

## 6 Conclusions

In global storm-resolving models, microphysical processes remain unresolved but in contrast to conventional climate models they are directly linked to the circulation that forces them. In this study we present a first exploration of microphysical sensitivities in a global storm-resolving model and examine the role that uncertainties in the representation of microphysical processes play for the tropical condensate distribution. We perform a microphysical ensemble with ICON that consists of eight simulations with a global 5-km grid and applies either a one-moment or a two-moment microphysics scheme. For the sensitivity

runs, we modify parameters of one hydrometeor category of the applied microphysics scheme. Microphysical sensitivities do not show strong situational or geographic variations, which gives us confidence that short runs that cover only a few days and global responses are informative to understand microphyscial effects.

We find that the two microphysics schemes have distinct signatures in how they distribute condensate among hydrometeor categories and differences are most prominent in the partitioning of frozen condensate. The one-moment scheme produces

overall less cloud ice, and more snow, than the two-moment scheme. These differences can be ascribed to the habit's definition

for each scheme which is associated with effective removal of ice at low burdens in the one-moment scheme and additional sinks for snow in the two-moment scheme. Overall differences between the simulations with the one-moment and two-moment schemes and differences resulting from parameter perturbations in a single scheme are moderate and tend to be larger for individual condensate habits than for more integrated quantities, like cloud fraction or total condensate burden.

Changes to the representation of one habit in a particular scheme generally affect different process rates of that condensate's habit but the implications of the modifications on a single property, the fall speed, largely determines the change in condensate amount: the faster a condensate falls, the less there is of it. Indirect changes of condensate amounts of the other habits tend to compensate for the change in the modified habit. Nevertheless, an increase in one condensate's fall speed also decreases total condensate burden. Even tropical mean precipitation efficiency is well explained by changes in the relative fall speed across different habits and both schemes. The importance of fall speeds across habits in this study agrees with evidence from earlier studies for specific habits (e.g. Sullivan et al. (2022) for ice fall speed, and Adams-Selin et al. (2023) for graupel and hail).

Given that the modifications of the sensitivity runs on condensate amount are moderate, the overall spread of a few W m$^{-2}$ at the top of the atmosphere is substantial and much larger than tropical mean day-to-day variability but smaller than from a first multimodel ensemble of global SRMs (global TOA: $\sigma$= 2.5 W m$^{-2}$ in this study; $\sigma$= 8 W m$^{-2}$ in DYAMOND from Hohenegger et al., 2020). Across schemes, perturbations of individual habits have consistent effects on the energy budget. In agreement with the robustness of the cloud fraction, changes in the radiative balance at the top of the atmosphere are dominated by changes in radiative properties of cloudy points rather than by changes in cloud fraction. In our simulations, only cloud water and cloud ice are seen by the applied radiation scheme, which plays a role in particular for simulations with the one-moment scheme that has much of its frozen condensate in the form of snow. Because frozen condensate tends to shift between hydrometeor categories rather than being added on top, rough estimates of the missed radiative impact lead us to believe that the current configuration rather overestimates the ensemble spread for the given parameter perturbations. It is an open question how much uncertainties in the representation of microphysics contribute to the multimodel spread of global SRMs and the design of our study does not allow us to directly approach that question. Taking into account that other studies indicate that sensitivities to horizontal and vertical model resolution are smaller than to parameterizations (Lang et al., 2023; Schmidt et al., 2024), this study points to a joint influence of the microphysics representation and other parameterizations, such as the turbulence scheme, and maybe even their interactions, to constrain uncertainties in global storm-resolving models.

The complexity of representing microphysics in numerical models has grown steadily, both in the design of bulk schemes (increasing number of habits and moments) and in the number of processes being represented, and it has been argued that the addition of uncertain parameters in more sophisticated schemes expands the space of possible cloud states (e.g., Morrison et al., 2020; Proske et al., 2022; Sullivan et al., 2022). On the counterpart, some efforts have been made to simplify the system while retaining its crucial properties (Wacker, 1995; Koren and Feingold, 2011; Mülmenstädt and Feingold, 2018; Proske et al., 2022). E.g., Proske et al. (2022) argue that few microphysical process rates dominate frozen and liquid condensate amount. In that sense, this study points to the importance of fall speeds for each of the considered habits. The simple relationship of less condensate for faster falling particles suggests that sedimentation is the dominant process while other conversion rates between habits that are affected by the fall speed play a minor role. One could adapt the view that the main question posed

for a microphysical scheme is how effectively it causes condensate to precipitate, either to the surface or to a level where it subsequently evaporates. From this perspective the different habits can be seen as ways to control condensate fall speeds, and hence total condensate amount. What makes a parameter effective in changing the behavior of the model may help design future sensitivity studies that more systematically explore how well one can fit representations of microphysical processes to their net effect on global radiation and the distribution of heating through the atmosphere.

*Code and data availability.* All simulations described in this study were performed on mistral at DKRZ using the ICON model, version icon-aes:icon-aes-two hashtag 0b9009842. The ICON source code, simulation runscripts, and additional python notebooks producing the figures of this manuscript are available in the related data repository (https://doi.org/10.17617/3.OD9NTK; Naumann, 2024). By downloading the ICON source code, the user accepts the license agreement.

## Appendix A: Modification of the microphysics parameterization

The specific modifications for the eight simulations are:

- **1mom**: one-moment scheme with standard parameters

- **2mom**: two-moment scheme with standard parameters

- **1mom-rain**: like 1mom but with a narrower raindrop size distribution. In the one-moment scheme, the raindrop size distribution, $n(D_r)$, is assumed to follow a gamma distribution in terms of the raindrop diameter, $D_r$:

$$n(D_r) = N_t N_0 D_r^\mu \exp\left(-\lambda D_r\right) \tag{A1}$$

  The standard parameters of $\mu = 0.5$ and $N_t = 0.1$ are changed to $\mu = 0.0$ and $N_t = 1.0$ for this run, which reduces the gamma distribution to an exponential distribution and corresponds to the classical Marshall and Palmer (1948) distribution.

- **1mom-ice**: like 1mom but with a higher ice fall speed. In the one-moment scheme, the fall speed of $q_i$ is given by

$$v_{q,i} = a(\rho q_i)^b (\rho_0/\rho)^c. \tag{A2}$$

  where $\rho$ is the air density and $\rho_0$ a reference surface air density. The standard parameter of $a = 1.25$, $b = 0.16$, and $c = 0.33$ are changed to $a = 3.29$, $b = 0.16$, and $c = 0.40$, where the modified $a$ and $b$ are suggested by Heymsfield and Donner (1990) and the modified $c$ is closer to the value given by Seifert and Beheng (2006).

- **1mom-snow**: like 1mom but with lower-density snow. Traditionally in microphysical schemes, that do not have a high-density frozen hydrometeor category like graupel or hail, as a compromise snow tends to be given higher density to allow for faster falling frozen particles. The applied one-moment scheme includes a graupel category and in this sensitivity

run, we make snow less dense, i.e., more distinct from graupel. To do so, we modify the snow particle's mass-diameter relationship, $m_s$-$D_s$, and the particle-based terminal fall velocity, $v_{p,s}$,

$$m_s = a_m D_s^2, \qquad\qquad v_{p,s} = v_0 D_s^\beta \qquad\qquad\qquad\qquad (A3)$$

by decreasing $a_m$ from its standard value 0.069 to 0.038 and decreasing $v_0$ from 25 to 20.

- **2mom-rain**: like 2mom but with a modified $\mu$-$D$ relationship. In the two-moment scheme, the raindrop size distribution is given by Eq. A1. While the product $N_t N_0$ and $\lambda$ are determined by the two prognostic variables $n_r$ and $q_r$, the shape parameter, $\mu$, is determined by a diagnostic relation of Seifert (2008) in the standard setup and changed to that of Milbrandt and Yau (2005) for the sensitivity run. For the sensitivity run, this implies a smaller shape parameter (i.e., wider RSD) for mean raindrop diameters $< 0.8$ mm and a larger shape parameter for mean raindrop diameters $> 0.8$ mm compared to the standard values. The variation in $\mu$ affects the fall speed of rainwater most strongly for relatively large mean raindrop diameters between 0.75 mm and 1.75 mm, where the diagnostic relation by Seifert (2008) assumes a local maximum in the fall speed of $q_r$ (Fig. 1 b). In this size range rainwater falls slower in terms of $q_r$ and faster in terms of $n_r$ in 2mom-rain, i.e., size sorting is weaker.

- **2mom-ice**: like 2mom but with a higher ice fall speed. In the two-moment scheme, the fall speed of cloud-ice particles, $v_{p,i}$, is given as a function of a cloud-ice particle's mass, $m_i$,

$$v_{p,i} = \alpha m_i^b. \qquad\qquad\qquad\qquad\qquad\qquad\qquad\qquad (A4)$$

We increase the standard value for $\alpha$ from 27.7 to 36.01. The fall velocities of $n_i$ and $q_i$ are obtained by integrating over the particle-size distribution and the change in $\alpha$ increases both fall velocities by about a third.

- **2mom-snow**: like 2mom but with modified snow properties. While the control run with the two-moment scheme applies snow properties suggested by Seifert and Beheng (2006), we apply the standard properties from COSMO for the sensitivity run. This implies modifying both the snow particle's diameter-mass relationship, $D_s$-$m_s$, and particle's fall speed, $v_{p,s}$,

$$D_s = a m_s^b, \qquad\qquad v_{p,s} = \alpha m_s^\beta \qquad\qquad\qquad\qquad (A5)$$

by decreasing $a$ from its standard value 5.13 to 2.4 and $b$ from 0.500 to 0.455, and by increasing the parameters $\alpha$ from 8.294 to 8.800 and $\beta$ from 0.125 to 0.150.

Although called the same (e.g., "snow") the morphology and properties assigned to each hydrometeor category (e.g., aggregate, diameter–fall-speed relation) are what defines the hydrometeor categories in each scheme. These assumptions differ between the specific one- and the two-moment scheme in this study, in particular for the frozen hydrometeors and therefore one must not expect them to agree. For example in the applied one-moment scheme, ice and snow are separated by size, while in the applied two-moment scheme they are separated conceptually in monomers and aggregates. In some aspects a comparison is nevertheless meaningful because of how they affect the energy and moisture budget.

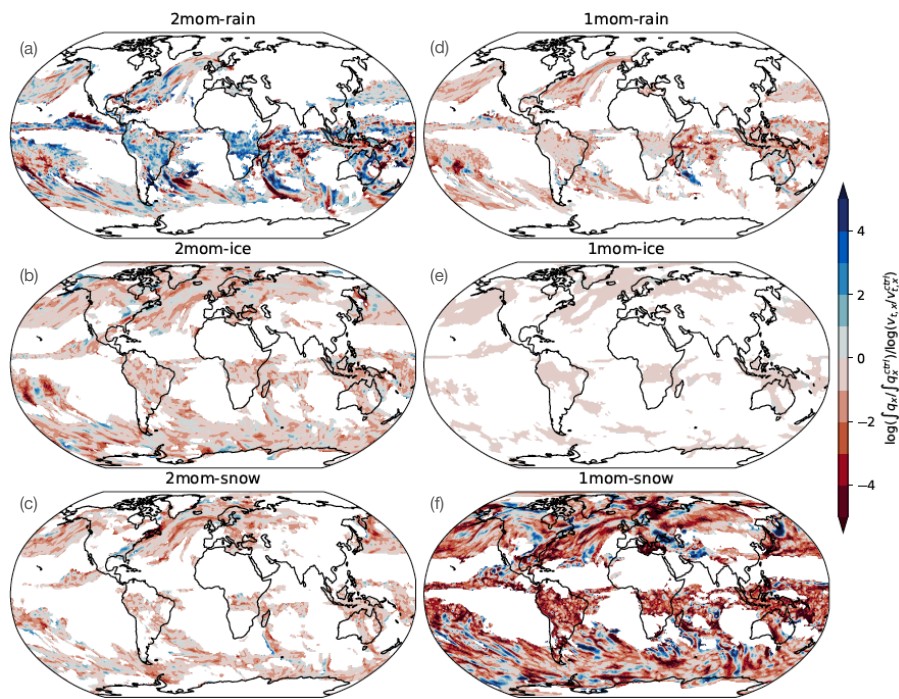

**Figure B1.** Global distribution of how condensate amount changes with a change in fall speed. We show the ratio of the logarithm of the normalized change in vertically integrated habit amount to the logarithm of the normalized change in that habit's fall speed, $\log(\int q_x / \int q_x^{\text{ctrl}}) / \log(v_{t,x}/v_{t,x}^{\text{ctrl}})$, for all sensitivy experiments, i.e., the slope between each of the simulations and their respective control run in log-space from Fig. 4 a. For simulations where the change in fall speed varies with $q_x$ or $m_x$, its average value is used. Regions where the considered condensate burden is low are masked. Negative values indicate that an increase in fall speed coincides with a reduction in condensate amount, or a decrease in fall speed coincides with an increase in condensate amount. E.g., a doubling of the fall speed while the condensate is reduced to half yields a value of -1.

## Appendix B: Regional distribution of relation between change in fall speed and change in condensate burden

The increase of condensate amount with decreasing fall speed per habit (and vice versa) as shown for the tropical mean (Fig. 4) also holds regionally in the tropics as well as globally (Fig. B1). The only exception is the 2mom-rain, which is, however, consistent with the regional ambiguity. The normalized overall signal is strongest for 1mom-snow in the tropics, somewhat similarly strong for 1mom-rain, 2mom-ice and 2mom-snow, and weakest for 1mom-ice.

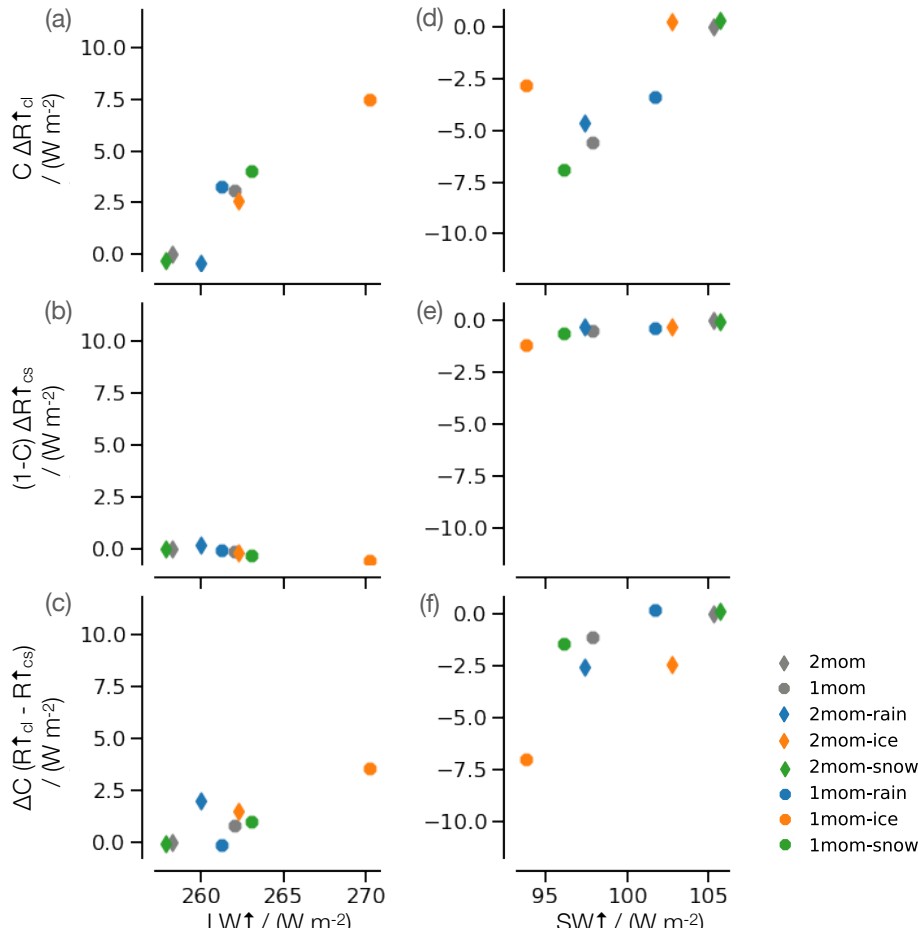

**Figure C1.** Decomposition of tropical radiative changes at TOA as difference to 2mom: (a,d) radiative change of cloudy points, (b,e) radiative change of clear-sky points, (c,f) radiative change due to cloud-cover change for (a-c) longwave and (d-f) shortwave radiation.

## Appendix C: Decomposition of tropical radiative changes at TOA

Differences in the TOA energy balance can be attributed to either changes in properties of cloudy points or clear-sky point, to changes in their percentage share (i.e., cloud cover change) or to higher-order terms:

$$\Delta R\uparrow \quad = \quad \Delta(CR\uparrow_{cl} + (1-C)R\uparrow_{cs}) \tag{C1}$$
$$= \quad C\Delta R\uparrow_{cl} + (1-C)\Delta R\uparrow_{cs} + \Delta C(R\uparrow_{cl} - R\uparrow_{cs}) + ... \tag{C2}$$

where $R\uparrow$ is either SW↑ or LW↑, $C$ is cloud cover, and the subscripts $_{cl}$ and $_{cs}$ indicate cloudy and clear-sky points, respectively.

The sum of the first three terms on the right hand side well approximates the total difference in the TOA energy balance so that higher order terms can be neglected.

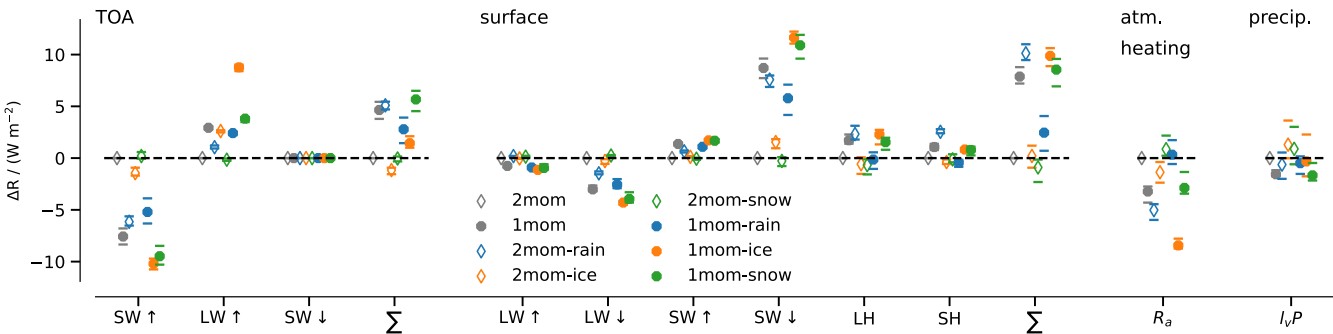

**Figure D1.** Same as Fig. 5 but for global means instead of the tropical belt.

The largest contribution to the total difference in the TOA energy balance is due to changes in the radiative properties of cloudy points in all simulations except 1mom-ice (Fig. C1). For an increased ice fall speed with the one-moment scheme, radiative changes of cloudy points and radiative changes due to a cloud-cover change contribute in similar magnitude to the total differences in the TOA energy balance. Compared to 1mom, 1mom-ice reflects less SW where cloudy points transition to clear sky but more SW where the ice concentration increases at cloudy point. In addition, outgoing LW radiation increases both at cloudy points due to a decrease in cloud-top height and where cloudy points transition to clear sky.

Changes in longwave properties of cloudy points are dominated by differences in the representation of ice-cloud microphysics in our simulations: lower and hence warmer cloud tops are consistent with an increase of LW↑ at TOA for cloudy points from 2mom to 1mom and with higher ice fall speeds. Changes in shortwave properties of cloudy points are affected both by ice-cloud properties and warm-cloud properties. For example, less frequent values of high cloud water for 2mom-rain decreases SW↑ of cloudy points compared to the other simulations with the two-moment scheme.

## Appendix D: Global mean energy balance

Contributions to the TOA and surface energy balance as well as atmospheric heating and surface precipitation are qualitatively similar for the tropical belt (Fig. 5) and globally (Fig. D1) but differences among simulations tend to be a bit smaller globally than in the tropics.

*Author contributions.* AKN developed the concept of the study and designed the experiments together with BS. ME implemented the two-moment scheme in this version of ICON with the help of AKN and performed the simulations. AKN wrote the code for the data processing, analysis and visualization. AKN and BS worked on the interpretation of the results. AKN wrote the manuscript with contributions from all co-authors.

*Competing interests.* The authors declare that they have no conflict of interest.

*Acknowledgements.* The authors thank Hauke Schmidt and four anonymous reviewers for helpful comments on earlier versions of the manuscript. AKN received funding from the Deutsche Forschungsgemeinschaft (DFG, German Research Foundation) under Germany's Excellence Strategy - EXC 2037 "Climate, Climatic Change, and Society" (project number 390683824). This project has further received funding from the European Union's Horizon 2020 research program under Grant Agreement No 101137680 via project CERTAINTY and Grant Agreement No 101003470 via the nextGEMS project. This work is also part of the WarmWorld project funded through the Bundesministerium für Bildung und Forschung (BMBF) and used resources of the Deutsches Klimarechenzentrum (DKRZ) granted by its Scientific Steering Committee (WLA) under project bm1183.

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
