# Peer review of "How the representation of microphysical processes affects tropical condensate in the global storm-resolving model ICON"

_EGUsphere, 2024_

## Referee Comment (RC3)

**Review of 'How the representation of microphysical processes affects tropical condensate in a global storm-resolving model'**

This manuscript investigates the impact of different representations of cloud microphysical processes on the distribution of tropical condensate in a global storm-resolving model using the ICON model. It focuses on the differences between a one-moment and a two-moment microphysical scheme and how modifying certain parameters within these schemes affects tropical-averaged species burdens, TOA and surface energy balance, and precipitation properties. Specifically, the change in the fall speed of hydrometeors significantly impacts both the distribution of condensate and precipitation efficiency, with faster fall speeds generally leading to less condensate and higher precipitation efficiency. Therefore, this manuscript may be worth publishing after addressing my concerns listed below:

**Minor Comments:**

The findings are specific to the ICON model and its configurations. The generalization of these results to other global SRMs should be made cautiously, as different models might exhibit different sensitivities to microphysical parameterizations. So, it is suggested to include "ICON" in the title for greater clarity.

While the manuscript is logically well-constructed in terms of scheme modifications, it solely focuses on model simulations and sensitivity studies but lacks direct comparisons with observational data. This omission limits the interpretation of the model's performance. It would be beneficial to include some case studies after the general assessment of the model changes, to demonstrate the efficacy of the different parameter perturbations.

The authors discuss the differences between the two schemes and the sensitivity tests, but a more elaborated discussion on the parameter-modified sensitivity runs is needed, given that these sensitivity simulations are one of the main focuses of the manuscript. For example, considering the discussions in Section 3, what are the corresponding implications of these parameter-induced changes in real-case simulation scenarios (which could be addressed in case studies)? Alternatively, how might these sensitivities impact the accuracy of climate predictions using the ICON model (considering a longer climatological simulation perspective)? I would like to leave the choice of such implementations to the authors.

Furthermore, the changes in species burdens appear to be determined by the changes in fall speeds, as suggested by comparing Fig. 4 and Fig. 1. However, it is difficult to quantify their sensitivities.

I suggest adding a quantitative assessment, such as evaluating the dependency of burden change rates on changes in fall speed. Additionally, it would be valuable to include the regional (latitudinal/longitudinal) distribution of these dependencies within the study domain in ICON.

**Minor Comments:**

**Line 31:** Please define NICAM.

**Section 2.1:** Does the ICON model's microphysical scheme used in this study include explicit aerosol-cloud interactions? If so, how are aerosols represented, what role do they play in the activation of cloud hydrometeors, and have these interactions been controlled across the different simulations?

**Section 2.2:** It would be helpful to include a table summarizing the sensitivity test configurations before discussing their potential effectiveness (e.g., in lines 115-116). This would provide a clear overview, and readers could be directed to the Appendix for more detailed information if necessary.

**Figure 1b:** Could you provide a potential explanation for the fluctuations in the 2mom standard rain falling speeds near the tails of the lines?

**Figure 2i:** The figure shows no hail mass across all simulations, yet hail fall speeds are presented for the 2mom standard simulation in Figure 1b. Could you clarify this discrepancy?

---

## Author Comment (AC1)

**Response to reviews on manuscript "How the representation of microphysical processes affects tropical condensate in a global storm-resolving model" by Naumann et al.**

*We thank the reviewers for their critical and constructive comments. In reading the reviews we better appreciated that we poorly communicated the motivation for our study, and that in the effort to communicate the more novel things we learned, some of the simpler aspects that would have reinforced our motivation got lost.*

*When beginning this work we were confronted with the daunting task of understanding how to tune the microphysical processes in a global cloud-resolving model. Questions we had were whether or not one need to simulate for a long time, we thought not, if the sensitivities would show strong situational or geographic variations from which it would be hard to draw conclusions, and if the simulations would be sensitive to the parameters we were familiar with as tuning parameters based on other studies. Given the computational intensity of the simulations, and the shear number of parameters, it was not possible to contemplate a systematic sweep of parameter space, and given the preliminary state of the model it seemed premature to embark on a fine tuning, with systematic comparisons to observations. Thus we decided to select a few parameters we were familiar with to address the above questions as a first exploration of how one would go about the tuning of our global model. Along the way we found that the sensitivity to the parameters is rather robust, it shows up right away and does not show substantial regional variation. In addition, it seemed to express a simple underlying principle, what we call the path to precipitation (i.e., how changing habit properties influence how fast they fall or how fast they become other habits that then fall), the communication of which motivated this paper. The fact that the study was based on more preliminary questions and that the paper was based more on what we unexpectedly discovered, the reader can rightfully be confused about the experimental design. In not drawing out more clearly some of the simple answers (short simulations are informative, and the sensitivities do not show large regional variations) the reader is also left to wonder if the results generalize.*

*We believe that by rewriting the introduction we could make this clear and thus help better frame what we did and what we learned. In the manuscript itself we would also give more emphasis to some of the little things we learned, if only as a way to emphasize how results generalize, and why the experimental design was as it was.*

*Based on this study, we do now have the confidence that short runs and global responses are informative, or rather that relatively little regional space needs to be covered in small domain case studies, to anticipate global responses. An idea of what makes a parameter effective in changing the behavior of the model will also help us design future sensitivity studies that will more systematically explore how well one can fit rather crude representations of microphysical processes, to their net effect on global radiation and the distribution of heating through the atmosphere.*

*In the following we respond to each of the reviewer's comments individually.*

**Reviewer 1**

The authors examine the role that uncertainties in the representation of microphysical processes play for the tropical condensate distribution in a global storm-resolving model. I agree with most of the conclusions but there are some different opinions about the comparison between single-moment and double-moment schemes, as I will outline further below. Therefore, I cannot recommend the publication of the manuscript at this stage.

Major comments:

1. I don't see any observations for comparison of model results. Thus, it is difficult to determine whether the modifications yield better results.

*We think the request for a comparison to observations stems from a poor communication of the manuscript's purpose. A five habit single moment scheme encapsulates many processes. If each processes is reversible, and interactions among habits are pairwise, plus interactions with water vapor, then 15 interactions are defined. We were more interested in trying to understand the main controls on behavior, rather than fine-tuning that behavior, as the latter also should be done holistically, along with the other processes, i.e., treatment of radiant energy transfer, or turbulent mixing. To address this point we modified the introduction to better explain our purpose and its motivation. E.g., we added: "Understanding how to tune the microphysical processes in a global SRM is a daunting task. Given the computational intensity of the simulations, and the shear number of parameters, it is not possible to contemplate a systematic sweep of parameter space, and given the preliminary state of the model it seems premature to embark on a fine tuning, with systematic comparisons to observations. Selecting a few parameters as a first exploration of how one would go about the tuning of a global SRM, [...]" Comparison of model results to observations for specific phenomena in ICON SRM have been taken up by other studies recently, e.g., for the structure of the ITCZ in the Pacific Ocean (Segura et al., in prep.).*

2. I agree with the discussion on single-moment or double-moment ensembles, but the discussion of the difference between single-moment and double-moment does not convince me. Whether or not to prognose the number of concentrations does not seem to be the only difference between the two schemes. Changing the description from single-moment to double-moment may make a big difference even only for one hydrometeor (e.g. cloud droplet in Xu et al 2024). Meanwhile, the difference in a single microphysical process scheme may also lead to large differences in water content, cloud cover, and precipitation (e.g. Lee et al). Therefore, if the authors want to compare the differences between single-moment and double-moment schemes, they need to ensure that the descriptions for other microphysical processes are consistent, for example, single-moment Thompson and double-moment Thompson schemes (e.g. Hill et al. 2015).

*It was not our intention to draw conclusions about one- vs. two-moment schemes in general, and we will revise the manuscript to avoid giving that impression. E.g., we have realised that parts of the introduction have been misleading in regard and clarify in the revised manuscript: "The two specific schemes applied in this study do not allow for general conclusions about the advantages of one type of scheme over the other, but they are rather interpreted as equally valid but different attempts to represent microphyscial processes in a global SRM." We do indeed compare a specific one-moment scheme with a specific two-moment scheme, and these two specific schemes differ in more than just the*

*number of moments but again, the intent was to see if we could find any order in this messy landscape. We believe we have, namely that everything revolves around the path to precipitation — i.e., how changes change the propensity of particles to fall, either by changing the rules that convert condensate into faster falling habits, by directly changing how fast a habit falls, or by indirectly influencing habit fall speed.*

Minor comments:

1. It is not a good reading experience to put the sensitivity test settings in the appendix. I recommend adding a table in the text to explain the settings of different tests. There is no reference for some test modifications (e.g. 1mom-snow, 2mom-ice).

*In the revised manuscript we add a table summarising the simulation setups in Section 2.2 of the main manuscript. The detailed description of the sensitivity experiments is, however, quite lengthy so that we prefer to keep it in the appendix. Not all test modifications that we apply have been published in peer-reviewed literature. E.g., 1mom-snow is based on an inconsistency in the applied assumptions that was present in the implementation of the one-moment scheme for a while.*

2. I agree with the importance of fall speeds, but the authors should at least give the differences in the microphysical processes affected by the fall speeds under different tests and draw the conclusion that "other conversion rates between habits that are affected by the fall speed play a minor role".

*For the sensitivity simulations, we modify parameters of the functional characterisation of the habits. These modifications affect a number of microphysical process rates of the affected habit, among them the habit's fall speed. Quantifying all affected process rates is hard to do without ambiguity because sometimes assumptions about fall speeds are hidden in other constants, or indirectly implicated to the extent one assumes geometric consistency. The main conclusion we take away from this study is that, despite using two schemes that differ from each other in their structure, and despite perturbing different parameters associated with different habits within these schemes, a large part of the effect of these parameter perturbations can be attributed to a single parameter, the fall speed. Hence we believe it is sensible to focus the method section (Section 2) on the effect of the modifications on the fall speed (Fig. 1), while we describe the specific modifications of the code in the appendix. We kindly ask the reader to refer to the comprehensive description of the schemes (Doms et al, 2018, for the one-moment scheme; Seifert and Beheng, 2006, for the two-moment scheme) for details of the formulation of the process rates.*

Hill, A. A., Shipway, B. J., & Boutle, I. A. 2015. How sensitive are aerosol-precipitation interactions to the warm rain representation? Journal of Advances in Modeling Earth Systems, 7, 987–1004. https://doi.org/10.1002/2014MS000422

Lee, H., & Baik, J.-J. 2017. A physically based autoconversion parameterization. Journal of the Atmospheric Sciences, 74(5), 1599–1616. https://doi.org/10.1175/jas-d-16-0207.1

Xu X, Heng Z, Li Y, Wang S, Li J, Wang Y, Chen J, Zhang P, Lu C. 2024. Improvement of cloud microphysical parameterization and its advantages in simulating precipitation along the Sichuan-Xizang Railway. Science China Earth Sciences, 67, https://doi.org/10.1007/s11430-023-1247-2

**Reviewer 2**

The article describes a set of experiments where parameters describing the habits of different cloud particle and hydrometeor types are varied. Their effects are evaluated in terms of differences between a one and a two moment scheme and the variations employed within them. This is done for a storm resolving model configuration, where the authors see extra impetus for their study because at that scale the dynamic forcing of cloud microphysics is resolved (convection), so uncertainties in the cloud microphysics are more important. The authors find that most of their variations relate to the fall speed of the hydrometeors, which they show correlates with the changes in condensate that they observe. They thus single out the fall speed an important tuning nob for the whole scheme.

I think this study includes strong ideas, for example the above-mentioned highlight of the fall speed importance, or the discussion around the precipitation efficiency, and is generally presented well. I discuss additional points below.

**Major**

To me the major weakness of the study seems to be that the experimental setup is not rigorous. For an exploration of the parameter space and its associated uncertainties, other studies have used setups thoroughly sampling the space. See for example Regayre et al. (2018) and references therein for the use of perturbed parameter ensembles for that purpose. This begs the question why you chose to sample only single parameter values?

*Because of the computational cost of simulations at global storm-resolving, i.e., kilometre-scale, grid spacing, we are limited in how many simulations we can run. E.g. the study by Regayre et al. uses a much coarser grid spacing of 1.25 x 1.875 deg. Indeed we are not aware of any study using a global storm-resolving model, which is able to run more than a few simulations. Hence, our motivation was less to understand what values of the parameters are correct, as we don't believe that any are. Bulk scheme are inconsistent under the most basic processes of transport and advection, so this type of modelling is really just about cutting's ones losses and capturing some basic effects. By varying the parameters following past practice in model tuning we mostly wanted to see if we could develop a conceptual framework, which is what we called the paths to precipitation. We revised the introduction and the method section to make this more clear.*

Also, for the sensitivity simulations it is important which parameters you choose to vary, by how much and why.

*We agree and we think that our work highlights how one might assess ahead of time what a change will do, i.e., through the metric it effectively influences the sedimentation.*

E.g. l. 102: why are you focusing on the functional characterization of the habits?

*This might reflect a different philosophical point of view. In the real worlds where we have real objects we can talk about processes. In the model the objects we deal with are defined by the rules that govern their behavior, rather than what we call them.*

l. 103-104: "what we deem a plausible range". How exactly did you pick that range? The information in Appendix A suggests e.g. comparison to what's been used in the literature. In addition, you are not sampling the range, but picking one value for each of the parameters (in addition to the default configuration). How did you decide on that value

after determining your plausible range (as opposed to e.g. sampling the minimum and maximum value)?

*Because none of the parameters are physical, they can't really be measured. We appreciate that we like to think of them physically, and by doing so we like to think of parameters that exist for physical objects as constraining the range of parameter values in the model. This means that the value of the parameters is subjective, and we just tried our best to pick what we thought is a reasonable range, mindful of the computational limitations our framework imposed upon us. In this we have been guided by literature values and what has been used in the past for tuning. The high resolution applied in our simulations strongly limits the number of simulation we were able to perform. Please see our response to the reviewers first comment.*

*For the analysis (e.g. Fig 4) we normalised by the relative magnitude of the perturbation (in terms of the fall speed), so that the result is independent of the magnitude as long as the response can be assumed to be close to linear. For other parts of the analysis (e.g. Fig 2), we were very carful not to compare the strength of two different parameter modifications (i.e., not to make statements like "perturbing ice has a stronger effect than perturbing rain" ) because this is indeed not something one can conclude from our study.*

l. 106: In explaining your choice it would help e.g. if you listed other parameters that could have been perturbed and that you decided against (because you think they'd have a smaller impact, as you say).

*Given the computational intensity of the simulations, and the shear number of parameters (we counted more than 200), it was not possible to contemplate a systematic sweep of parameter space. We decided to focus on the properties that define the habits in the bulk schemes, rather than parameters that determine the process rates. We clarified this to the discussion in Section 2.2.*

From Appendix A I gather that at least for some of the perturbations you had physical properties in mind, for example affecting the size sorting. In my understanding these are unconstrained properties of the hydrometeors that you deem as sources for uncertainty, because the choice for the corresponding values is underdetermined. I think you should highlight that more in the introduction to justify your choice of parameters. For example you could have a table for the different sensitivity simulations, with a column explaining the target of the sensitivity simulation (e.g. making the snow less dense and more distinct from graupel in 1mom-snow).

*We revised the introduction to clarify this point and added a table summarising the simulation setups in Section 2.2.*

For some parameters your choice for the values is justified well (e.g. for the 1mom-rain distribution changing from gamma to Marshall and Palmer), but for others it is not well explained and thus seems random (e.g. for 1mom-ice you choose parameters from different publications, and for c not even the one from the publication but one that's closer to it; for 1mom-snow there is no justification at all for the specific value). Here you could elaborate your reasoning or state that it was chosen ad-hoc and what caveats for your interpretation of the results it brings with it.

*Please refer to our overall comment at the beginning of this document. We revised the introduction and method section to make our reasoning more clear.*

l. 131: You state yourself that the parameter perturbations were conservatively chosen. Why do you infer that from the comparison to the one- to two-moment scheme difference (e.g. should that be smaller for physical reasons or do literature studies suggest that)? As stated above, I think you also need to make more clear why you choose the parameter values "conservatively", rather than e.g. using the minimum and maximum value of your plausible range.

*To the degree that the two schemes (with their default values) are sensible representations of microphysical processes and their difference is representative for our uncertainty in representing the system, the fact that differences between simulations with the one-moment and two-moment scheme tend to be larger than differences resulting from parameter perturbations in a single scheme, make our choice of parameter perturbations conservative.*

In general, in the introduction as well as in the discussion of the results, the study could benefit from more reference to literature on the impact of unconstrained choices in microphysics schemes. For example , the group of Adele Igel has produced a lot of interesting work on the topic, see e.g. Hu and Igel (2023).

*Thanks, we agree, given the comments by the reviewer we appreciate that this context would be useful, even if previous global studies were using very different modelling approaches (i.e., parameterized vertical motion), or focus on much smaller or idealised domains (like Hu and Igel, 2023). We added this discussion in the revised manuscript.*

In relation to such literature and to your particular study setup, the particular question you are trying to answer remains unclear for me. You are sparsely and conservatively sampling the parameter space, so it is not a thorough exploration of uncertainty. Your are not comparing between resolutions (side note: that would be interesting to add), so you cannot answer whether the change in resolution or in parameters/schemes produces more variance. You are not attempting to tune your model. Is it just a first test of cloud microphysical influences in the storm resolving ICON version? I would then advise to state that more clearly.

*Please see our overall comment in the beginning of the document. We revised the introduction accordingly to clarify this point. E.g., in the revised manuscript we state this now more clearly already at the end of the first paragraph in the introduction: " This motivates the present study, which presents an exploration of microphysical sensitivities in the first global storm-resolving version of ICON (ICOsahedral Nonhydrostatic model; Hohenegger et al., 2023) […] ." And later: "Understanding how to tune the microphysical processes in a global SRM is a daunting task. Given the computational intensity of the simulations, and the shear number of parameters, it is not possible to contemplate a systematic sweep of parameter space, and given the preliminary state of the model it seems premature to embark on a fine tuning, with systematic comparisons to observations. Selecting a few parameters as a first exploration of how one would go about the tuning of a global SRM, […]"*

One key results is that you see that changes in condensate can be attributed mainly to a change in fall speed. However, since your perturbations are chosen in a peculiar way, one wonders how much this result is due to your setup? So does fall speed only come out as important because that is the property that you perturb the most?
Here it could help to add to your discussion in l. 111 - 115 with a description of which

processes the habit parameters you change affect directly (for example with a graphic). Thus one could judge better which processes even have a chance to compete with the fall speed for importance with the perturbations.

*We might have been lucky, or unlucky, that all of our changes mapped on to fall speed, but we think there is reason to believe that this is a useful organizing principle — essentially our hypothesis — rather then a very lucky choice of which parameters to vary. Please also refer to our response to reviewer 1's minor comment 2.*

Do you have a similar comparison as Fig. 4 also for the other processes that were perturbed?

*We are not aware of any other process than the fall speed that we perturbed in similarly consistent manner across the simulations or a subset of them.*

What do you suggest how we continue from here?

*Our future work is going in two directions, one to use schemes which are asymptotically consistent, i.e., super-particles as they approach the physical system with enough degrees of freedom. The other is to explore the behavior of yet simpler schemes, with fewer (or more diagnostic) habits, where the processes that govern how precipitate is formed and falls can be more functionally constrained.*

Your introduction opens up the considerations for choosing between one- and two-moment schemes. Can you add to those considerations?

*Please refer to our response to reviewer 1's second major comment.*

(How) Should we go about exploring parameter uncertainties more thoroughly? In l.108-110 you say that all simulations are equally likely. What does that range mean for interpreting single km-scale simulations as it is often done, or how could you further explore that spread? Do you have ideas for how to constrain the fall speed and related parameters?

*These are good questions. For the use of the 1 and 2 moment schemes our present approach would be to do what the reviewers hoped we would have done for this study, which is to use observations. For this we now have the new EarthCARE measurements together with coordinated curtains made with HALO during the ORCESTRA campaign. Based on these we will focus on controlling the total condensate burdens so that we have a similar radiative signature as in the observations. This work is being informed by the present study, as the number of parameters in the schemes is very large (we counted more than 200), thus we would use the insights from this study help guide our choices.*

l. 87 - 91: You are comparing the global mean differences between your sensitivity simulations to the global mean differences between different days of the simulation. I agree that this shows some robustness of the sensitivity simulation differences, but it's not clear to me that this is a reasonable metric for comparison. Why not compare to variability in space or shorter time intervals instead?

*In the manuscript we focus on tropical means (e.g., Fig. 2) or statistics aggregated over the tropical belt (e.g., Fig. 3). To assess the robustness of the differences between simulations it therefore seems reasonable to focus on tropical means or tropical aggregated statistics instead of regional variability. To use the difference between days is a*

*practical choice given the length of the simulation that avoids the need to consider effects of the diurnal cycle.*

**Minor**

- l. 4-5: "where we modify parameters of one hydrometeor category of the applied microphysics scheme": add "in each" or so to clarify that each sensitivity simulation modifies one parameter in one category.

*Done.*

- l. 6: "and can be ascribed..." this second part of the sentence is unclear to me.

*We clarify: "and can be understood in terms of the habit's definition…"*

- l. 7: moderate compared to what?

*Compared to inter-model differences show, e.g., by Roh et al. 2021, or to regional variability.*

- l. 19: "resolved away" sounds strange to me, rephrase perhaps?

*We rephrased: "disappear even if the trend for increasing resolution continues"*

- l. 22: give some examples of those past studies (at least with citations)

*We added citations here.*

- l. 31: explain NICAM abbreviation

*Done.*

- l. 36: do you mean at km-scale horizontal resolution it's not so important which one one chooses (e.g. 10, 5, or 2 km)? Or going to km-scale resolution at all is less important compared to the schemes?

*The first, i.e., horizontal resolution at km-scale range. We clarify that in the manuscript.*

- l. 45: allow*s*

*Done.*

- l. 45-51: I value your explanation of the considerations going into using a two-moment scheme or not. You are weighing more flexibility and number information against complexity and computational burden and conclude that the advantage is unclear. But I don't see the connection to your study, where you are comparing results of the one- and two-moment scheme (and quote the additional computational burden). Thus I think this part of the introduction should explain why you have reason to believe that the two-moment scheme gives different/better results (taking up the point from above you could e.g. on past studies comparing results). Or else take these points up in your results and discussion more explicitly, elaborating what e.g. the additional number information has done, or how complexity got in the way, etc. Or maybe your standpoint is that one can justify either choice (for one- or two-moment scheme), and that therefore you consider it a

source for uncertainty and treat both. All these alternative argumentation avenues may be implicit for you, but they are not clear for me as a reader.

*Please see our response to reviewer 1's second major comment.*

- l. 53: You refer to other studies' findings for the tropics in the first paragraph of that page, but what exactly is your motivation for focusing on the tropics?

*The advantage of global SRMs is argued to be largest in the tropics where convection is often directly driven by diabetic processes rather than by larger dynamical systems like low pressure systems in the mid-latitudes. We believe this is also why other SRM studies often focus on the tropics.*

- l. 85: why restrict the analysis to 5 of the 10 days?

*We clarify in the revised manuscript: "To allow for ample time for spinup, the analysis shown in this study is restricted to the last 5 days of these 10-day simulations…".*

- l. 85: I think the Figure captions should still say explicitly that they are restricted to the tropics.

*Agreed. For Figure 3 this information was indeed missing and we added in the revised manuscript.*

- l. 87: give some example citations

*We added: "Xue et al. (2017) for a squall line case, or VanZanten et al (2011) for shallow cumulus"*

- l. 140: allow*s*

*Done.*

- l. 141-146: For a reader not knowing these schemes perhaps a graphic showing the processes included in each of them could help (in combination with how the habit parameters that you adjust impact these processes).

*Please see our response to reviewer 1's minor comment 2.*

- l. 155: Are these results shown anywhere?

*Yes, in Fig 4 as stated in the beginning of the paragraph.*

- l. 167: Is this a hypothesis or have you checked that e.g. with analysing process rates?

*As one would expect from process understanding, the formulation of the process rate in the applied one-moment scheme indeed shows that evaporation from rain water increases with the amount of available rain water.*

- l. 179: I don't see where you show that for condensate amount. On pg. 5 it's also only justified with Fig. 5 and C1, which relate to the energy balance.

*In the first paragraph of Section 3, we discuss the condensate amount and referring to Fig. 2 write that "Differences between the one-moment and two-moment scheme tend to be larger than differences resulting from parameter perturbations in a single scheme…".*

- l. 190: Point to Fig. B1

*Upon revisiting the paragraph, we realised that the content of the sentence the reviewer refers to occurred twice in the same paragraph. We therefore reformulate this part and point to Fig. B1 later in the same paragraph.*

- Fig. B1: caption d-f: shortwave

*Done.*

- l. 192: The wording here was confusing to me, e.g. "high-cloud fraction is just a little higher".

*We clarified to "… high-cloud fraction is just a little larger…"*

- l. 206: why is it "probably small"?

*For clarity we added: "… due to the tendency of SW$\uparrow$ and LW$\uparrow$ to compensate each other"*

- l. 242: where is the information on the evaporation?

*We added: "(see discussion in Section 3.2)"*

- l. 277: But you also say your perturbations are conservative. So the spread would be underestimated overall?

*For clarity we add: "… for the given parameter perturbations."*

- l. 281: What do you mean by "common"?

*Here we mean "common" in the sense of "joint" rather than "usual". We clarify by replacing "common" with "joint".*

- Figure legends throughout: Why not order the legend by 2mom and then 1mom so that e.g. in Fig. 5 the colours line up? Also, I find the symbols hard to distinguish. Could you use something more different than circle and diamond?

*We tried different possibilities but still find it most clear to group the simulations with default values (i.e., 1mom with 2mom). Also, we will pay attention that figures are shown large enough so that symbols are easy to distinguish in the next version of the manuscript.*

**References**

Hu, Arthur Z., and Adele L. Igel. "A Bin and a Bulk Microphysics Scheme Can Be More Alike Than Two Bin Schemes." Journal of Advances in Modeling Earth Systems 15, no. 3 (March 2023): e2022MS003303. https://doi.org/10.1029/2022MS003303.

Regayre, Leighton A., Jill S. Johnson, Masaru Yoshioka, Kirsty J. Pringle, David M. H. Sexton, Ben B. B. Booth, Lindsay A. Lee, Nicolas Bellouin, and Kenneth S. Carslaw. "Aerosol and Physical Atmosphere Model Parameters Are Both Important Sources of Uncertainty in Aerosol ERF." Atmospheric Chemistry and Physics 18, no. 13 (July 13, 2018): 9975–10006. https://doi.org/10.5194/acp-18-9975-2018.

**Reviewer 3**

This manuscript investigates the impact of different representations of cloud microphysical processes on the distribution of tropical condensate in a global storm-resolving model using the ICON model. It focuses on the differences between a one-moment and a two-moment microphysical scheme and how modifying certain parameters within these schemes affects tropical-averaged species burdens, TOA and surface energy balance, and precipitation properties. Specifically, the change in the fall speed of hydrometeors significantly impacts both the distribution of condensate and precipitation efficiency, with faster fall speeds generally leading to less condensate and higher precipitation efficiency. Therefore, this manuscript may be worth publishing after addressing my concerns listed below:

Minor Comments:

The findings are specific to the ICON model and its configurations. The generalization of these results to other global SRMs should be made cautiously, as different models might exhibit different sensitivities to microphysical parameterizations. So, it is suggested to include "ICON" in the title for greater clarity.

*We changed the title to: 'How the representation of microphysical processes affects tropical condensate in the global storm-resolving model ICON'*

While the manuscript is logically well-constructed in terms of scheme modifications, it solely focuses on model simulations and sensitivity studies but lacks direct comparisons with observational data. This omission limits the interpretation of the model's performance. It would be beneficial to include some case studies after the general assessment of the model changes, to demonstrate the efficacy of the different parameter perturbations.

*Concerning the comparison to observations please refer to our overall comment at the beginning of this document and our response to reviewer 1's major comment 1. We were indeed interested if sensitivities would show strong situational or geographic variations but found that they do not vary substantially on the regional scale. We add a discussion on this in the new appendix B.*

The authors discuss the differences between the two schemes and the sensitivity tests, but a more elaborated discussion on the parameter-modified sensitivity runs is needed, given that these sensitivity simulations are one of the main focuses of the manuscript. For example, considering the discussions in Section 3, what are the corresponding implications of these parameter-induced changes in real-case simulation scenarios (which could be addressed in case studies)? Alternatively, how might these sensitivities impact the accuracy of climate predictions using the ICON model (considering a longer climatological simulation perspective)? I would like to leave the choice of such implementations to the authors.

*How theses sensitivities map to the climate time scale is an interesting question to explore in future studies. We believe that this studies gives us an idea of what makes a parameter effective in changing the behavior of the model which will also help us design future sensitivity studies that explore their effect on climate predictions.*

Furthermore, the changes in species burdens appear to be determined by the changes in fall speeds, as suggested by comparing Fig. 4 and Fig. 1. However, it is difficult to quantify their sensitivities. I suggest adding a quantitative assessment, such as evaluating the dependency of burden change rates on changes in fall speed. Additionally, it would be valuable to include the regional (latitudinal/longitudinal) distribution of these dependencies within the study domain in ICON.

*In the revised manuscripts we try to give a more quantitative sense of this, but would also like to avoid false precision. The main point is that variations in fall speed, or the rate at which species transition to other species with larger fall speeds, is the main control on condensate burden. This also seems to work across various regional regions. To make this more clear in the revised manuscript we add quantitative global map plots of how condensate amount changes with a change in fall speed in the new Appendix B.*

Minor Comments:

Line 31: Please define NICAM.

*Done.*

Section 2.1: Does the ICON model's microphysical scheme used in this study include explicit aerosol-cloud interactions? If so, how are aerosols represented, what role do they play in the activation of cloud hydrometeors, and have these interactions been controlled across the different simulations?

*We add this information in Section 2.2.: "In both schemes, a constant concentration of cloud condensation nuclei is prescribed."*

Section 2.2: It would be helpful to include a table summarizing the sensitivity test configurations before discussing their potential effectiveness (e.g., in lines 115-116). This would provide a clear overview, and readers could be directed to the Appendix for more detailed information if necessary.

*We add a table summarising the simulation setups in Section 2.2.*

Figure 1b: Could you provide a potential explanation for the fluctuations in the 2mom standard rain falling speeds near the tails of the lines?

*We believe the reviewer is referring to the local maximum in $v\_q$ and the local minimum in $v\_n$ at around m=10-6 kg. These extrema are related to the mu-D relationship applied in 2mom by Seifert (2008; Fig. 6 in that study), which assumes a minimum shape parameter, mu, for a mean raindrop diameter of 1.1 mm. Seifert (2008) argues that at this raindrop size mu reaches a minimum (i.e. a broad raindrop size distribution) which is associated with the precipitation maximum of a convective event. The implication of such a broad raindrop size distribution is a local maximum in $v\_q$ (and minimum in $v\_n$). In the revised manuscript, we add in Appendix A for 2mom-rain: "The variation in mu affects the fall speed of rainwater most strongly for relatively large mean raindrop diameters between 0.75 mm and 1.75 mm, where the diagnostic relation by Seifert (2008) assumes a local maximum in the fall speed of $q\_r$ (Fig. 1 b)."*

Figure 2i: The figure shows no hail mass across all simulations, yet hail fall speeds are presented for the 2mom standard simulation in Figure 1b. Could you clarify this discrepancy?

*As a hydrometeor category hail is only present in the two-moment scheme, not in the one-moment scheme. Therefore hail fall speeds are presented in Fig. 1 b but not in Fig. 1 a. In addition with the two-moment scheme tropical mean specific mass of hail is orders of magnitude smaller than the specific masses of the other habits, which is why (plotted on the same y-scale for comparability) hail appears as a straight line in Fig. 2 l.*

**Reviewer 4**

Overall, this is an interesting study that explores the sensitivity of global storm resolving models that resolve deep convection to the representation of microphysical processes. This research could be a valuable contribution to the literature as these types of sensitivity studies have not been explored in much detail yet. However, I share the concerns of the other reviewers that this study lacks rigor in terms of the choices made to compare the two different microphysics schemes and in terms of the parameter choices that were made. Given the different process representation in the one and two moment schemes and the lack of rigor in the variation of the parameter choices, it is hard to tell how rigorous the conclusion that the change in fall speeds is most significant for the representation of cloud microphysical processes. In addition to concerns about the methodology of the study and analysis of results, the presentation of the research throughout the paper could be significantly improved. The figures were not very well thought out and often quite difficult to interpret. Notation is often defined only in the figure caption and not fully explained, and should be included in the main text of the paper instead.

Specific Comments:

Lines 22 – 24. The sensitivity to cloud microphysics in coarse climate models vs. global storm resolving models was given as a motivation for this study, but there's no real discussion anywhere in the paper of the results of this study in the context of how these results compare against coarser scale climate models and their sensitivity to cloud microphysics.

*Thanks, we agree, given the comments by the reviewer we appreciate that this context would be useful, even if previous global studies were using very different modelling approaches (i.e., parameterized vertical motion), or focus on much smaller or idealised domain. We added this discussion in the revised manuscript.*

Lines 25-27. Is there really considerable diversity in the representation of cloud microphysical models across the DYAMOND simulations? Cloud microphysical schemes in these models are limited to one or two moment bulk cloud parameterizations which have many known limitations.

*To our knowledge that all of the simulations in the DYAMOND ensemble apply a one-moment scheme. They are mostly inherited from their "parent" conventional (CMIP-style) climate model, so we would expect a similar diversity in the representation of cloud microphysics across DYAMOND simulations as across CMIP simulations. Also, different studies argued that different types of microphysics schemes (one-moment vs two-moment vs bin) do not necessarily give results that are more different than within one type (e.g. VanZanten et al. 2011, Morrison et al. 2020, Hu and Igel 2023).*

Lines 50-51. There have been a number of studies that have demonstrated the advantages of multi-moment schemes over single moment schemes. It's also worth pointing out here that there is physical uncertainty in many of the microphysical processes.

*In the revised scheme we add in the introduction: "While a number of studies have demonstrated the advantage of multi-moment schemes over one-moment schemes often in regionally constrained case studies and related to the representation of specific processes (e.g. Sullivan et al., 2023; Seiki et al., 2015 for a global domain), the situation is less clear for global or more aggregated statistics relevant for climate studies and some*

*efforts have been made to identify processes, or process groups, that dominate the system across schemes (Wacker 1995, Koren and Feingold 2011, Muelmenstaedt and Feingold 2018, Proske et al. 2022)." Concerning the reviewer's second point, we write in the first sentence of the same paragraph in the introduction: "Uncertainties in the microphysical parameterizations used in global SRMs can arise due to the basic approaches employed, or from uncertainty in how to represent specific processes within a specific approach, or due to limited understanding of the process itself. "*

Lines 81-86. Are there potential biases in terms of using simulations that focus on a two-week time-period during Northern Hemisphere winter?

*The sensitivities we explore in this study do not show strong situational or geographic variations (see new Appendix B), which gives us confidence that short global runs are indeed informative, and do not depend on the specific period chosen.*

Section 2.2. The description of the sensitivity experiments should be included in the main text, not as part of the appendix. It makes it very hard to follow or understand the rest of the paper without these descriptions being in the main text.

*As also suggested by reviewer 1, we add a table summarising the simulation setups in Section 2.2 of the main text. The detailed description of the sensitivity experiments is, however, quite lengthy so that we prefer to keep it in the appendix.*

It would be useful to have a figure or table illustrating the different microphysical processes that are accounted for in both schemes. Given that the differences between the schemes are not solely due to using a one moment or two moment representation for the droplet or particle size distributions, it would be easier for the reader to follow the comparison between the different simulations if it is clear what process rates are included in each scheme.

*Please refer to our response to reviewer 1's minor comment 2. We discuss fundamental differences in process rates between the two scheme that are relevant for specific differences in the simulations, in particular the snow-ice partitioning being related to representation of sinks and sources of ice and snow in Section 3.1.*

Lines 105-107. More justification is needed for the choice of plausible parameter values based on past literature.

*Please refer to our overall comment in the beginning of this document and our response to reviewer 2's fourth major comment.*

Lines 107-110. Was there any comparison done against observations?

*No, we do not compare to observation in this study. Please see our overall comment in the beginning of this document and our response to reviewer 1's major comment 1.*

Figure 1. Why is there a slash in the x and y labels between the variable name and the units? I think this is supposed to the denote the units in some way but this not conventional. The label should describe what is shown in words (e.g. mass mixing ratio or number concentration), not use notation that is not clearly defined. The right-hand figure would also be clearer if the number and mass were plotted on different y-axis (left and right for example). I also don't clearly understand what is shown here. It's not clear when

what is shown represents the "typical" one and two moment schemes or those modified for the sensitivity experiments.

*The slash between variable name and the units follows the quantity calculus of Taylor (2018, https://nvlpubs.nist.gov/nistpubs/jres/123/jres.123.008.pdf), and as such is not uncommon in the field. One motivation is that the space the graphs is unitless so the quantity calculus shows what we actually present, which is a quantity divided by its unit. All symbols that are used to label the axis are clearly defined in the figure caption. We think that it is exactly that precision that accompanies the introduction of a symbols which is why they should be favored in a figure.*

*Both panels show fall speeds. In the one-moment scheme (panel a), we only have the fall speed of the specific mass, $v_q$. Different habits are shown in different colours as indicated by the legend. Thick lines represent the default values of the one-moment scheme used in this study for the control runs (i.e., 1mom), while thin lines represent the modified fall speed of a particular habit in a particular sensitivity run (e.g., thin blue line for the modified rain fall speed in 1mom-rain). The two-moment scheme distinguishes between the fall speed of the specific mass, $v_q$, and the fall speed of the specific number, $v_n$ (panel b). However, they are both fall speeds which is why we think it is an imperative to plot them on the same y-axis. Colors and line thickness are used as in panel a. We clarify the use of colours and line thickness in the figure caption of the revised version of the manuscript.*

Lines 124. What is a trimodal cloud fraction structure?

*With trimodal cloud fraction structure we refer to the three maxima in the cloud fraction profile (shallow, mid-level, and high convection). We clarify that in the revised manuscript.*

Figure 2. Again, the notation and labels here do not follow conventional notation and are very unclear and hard to understand. The color scheme is also challenging to read. What is specific water vapor at difference to 2mom? Is this the default 2mom, and how is it defined? Why is it shown here in the overview figure of the ensemble rather than as a separate figure? There's barely any explanation and discussion of this figure in the text so the choices that are made are frankly confusing. Since the main result discussed in the text here is the difference between the snow and ice amounts between the one and two moment schemes, why not have a figure that emphasizes only those 2 panels?

*Concerning the figure design please see our response to the reviewer's comment on Figure 1 above. For specific water vapour we show the difference to the simulation named "2mom" because in absolute values no difference would be visible in the figure. We also think it is worth to show all the panels to get a good overview of the simulations. Although we discuss variables that differ more strongly among the simulations (e.g. ice and snow) in more detail, it is also an important result, and noteworthy to show, that the simulations differ less in other variables.*

Since the main difference between the one and two moment schemes seems to be associated with the snow and ice amounts, it would make sense to include a sensitivity experiment where the auto-conversion threshold between ice and snow is changed in the one moment scheme. How does the typical snow-ice auto-conversion threshold differ in the default simulations between the one and two moment schemes?

*We ran small test simulation to better understand why there is so little ice in simulations with the one-moment scheme, and found that the formulation of autoconversion and*

[Figure]

Vertical profiles avg. over the tropics

Ice: solid, Snow: dashed

*Fig 1: Profiles of ice (solid lines) and snow (dashed lines) in test simulation with the one-moment scheme. Compared to the default parameters ("all process ON"), turning off deposition-autoconversion ("Deposition OFF") is more important for increasing ice amounts than autoconversion and accretion ("Ice2snow OFF"). Figure courtesy: Romain Fievet*

*accretion, which turns ice into snow ("ice2snow" in Fig. 1 in this document), both play a minor role. Instead the deposition-autoconversion (i.e., the conversion from vapor plus ice to snow; "deposition" in Fig. 1) is the dominant process that depletes ice within the one-moment scheme. In the particular two-moment scheme used in this study ice and snow are differentiated by morphology (monomers versus aggregates) and hence deposition-autoconversion as a process is not considered. We discuss this in Section 3.1 in the manuscript but without the reference to the test simulations. Running an additional sensitivity experiment (either on autoconversion or deposition-autoconversion) with the manuscript's global storm-resolving setup is unfortunately to costly.*

It's also unclear how much difference in cloud cover between the two schemes plays a role in the interpretation of Figure 2, and potential feedbacks between cloud microphysics and macrophysics were not discussed in the text in this section. Since all of the metrics are aggregated as global or regional mean profiles, it's not clear whether these types of effects would be clear from this analysis.

*We are not sure if we fully get the reviewer's questions here. In Fig. 3 we show spatial variability in response to microphysical perturbations which is consistent with global profiles in Figure 2. In the revised version we added a regional analysis in Appendix B.*

Figure 4. Again, the notation and labels should be described in words rather than using notation. If the control in each case refers to the default 1 moment or default 2 moment scheme, it's really not clear to put these both on the same plot, since the normalization means two different things in this case. The choice of axis ranges is also quite strange in this figure; why are the ticks not evenly spaced or labeled in this figure? It's really hard to understand why these choices were made for the presentation of the data here, and the explanation in the text gives no explanation either.

*Concerning the figure design please see our response to the reviewer's comment on Figure 1 above. The difference of each sensitivity simulation from either 1mom or 2mom can be characterised by a change in fall speed of one of the hydrometeor categories. We therefore think it is consistent to show the change in fall speed (x axis) and condensate amount (y axis) relative to the simulation it differs from (i.e., either 1mom or 2mom) and that this kind of normalisation indeed "means the same thing". All variables are plotted on a logarithmic axis so that e.g. a doubling and a halving of the fall speed or the condensate amount shows up as the same size of an effect. We clarify the use of a logarithmic axis in the figure caption of the revised version of the manuscript.*

Figure 5. This figure needs to be larger, and I also think it would be easier to interpret if the TOA and surface were plotted as different figures, rather than a single continuous figure, as Epsilon here means two different things. What do the error bars represent here? It would also be helpful to more clearly differentiate between the 1 and 2 moment schemes (using for example filled or empty markers, rather than just differences in shape). Since the sensitivity experiments between 1 moment and 2 moment are not directly comparable, it also doesn't really make sense to use the same colors as though they are directly comparable. What is $l_v P$ ?

*In the revised version of the manuscript we will make sure that the figure is printed larger and we clarify that "error bars indicate the range from the minimum to the maximum daily mean within the 5-day period". We also add that l_v P is the latent heat of vaporisation times surface precipitation. However, we believe that the separation between TOA and surface is already very clear and that both sigmas (which we believe is what the reviewer means by "epsilon") are clearly attributable to either one of them.*

Figure 6. This figure is also too small.

*We rearranged the panels so that the figure can be enlarged in the revised version of the manuscript.*

Conclusions. It would be useful to discuss how the results for global storm resolving models compare against the sensitivity of cloud microphysics in coarser climate models. Other recent studies have also pointed to the important of ice fall speeds or ice-snow auto-conversion in perturbed parameter ensembles of CAM6 [Duffy et al. 2024] and it would be useful to cite these results here in the context of the GSRM sensitivity study.

*Thanks. We added this in the revised version of the conclusions.*

*Taylor, B. (2018): Quantity calculus, fundamental constants, and SI units, Journal of Research of the National Institute of Standards and Technology, Volume 123, Article No. 123008, https://doi.org/10.6028/jres.123.008.*

---

## Author Response (AR2)

**Response to reviewer comments regarding egusphere-2024-2268:** 'How the representation of microphysical processes affects tropical condensate in the global storm-resolving model ICON'

We thank the editor and the reviewers for the additional time they devoted to the manuscript, and are glad we were better able to articulate the scope and motivation for the manuscript. We have addressed the comments raised in reports 2 and 3 through changes to the manuscript as noted below.

**Report 2:** raised the following points, our responses follow in color:

1. relation of the work to Mauritsen et al.: "This discussion seems to be closely linked to Mauritsen et al. (2022)"
2. clarification in the text: l. 67: unclear what "they" refers to and l. 68: investigate (no s), and l. 41-43: "this comparison is still not clear to me"

We have now slightly restructured the introduction to present the work in light of the work of Mauritsen et al., which we also now cite. The sentence with the unclear 'they' has been rewritten to avoid too many indirect references. The typo (investigate) has been corrected. The sentence with the confusing reference to the comparison to grid spacing has been removed. It was not necessary and was presented more clearly elsewhere.

**Report 3:** raised three issues, which we address in turn (in color):

Lines 61 – 68. It seems like the connection between changing habit properties and fall speeds could have been inferred using a much less computationally intensive model. Thus it's still very unclear what the value of doing this type of sensitivity study in a global SRM is. It seems like in future work the parameter spread could first be evaluated in a much less computationally expensive model and then this could be used to determine reasonable parameter values for the much more computationally expensive GSRM. Why was such an approach not taken here?

Yes in principle one could do as the reviewer suggested, but it leaves open the question as to whether the sensitivities that are identified in simple settings manifest themselves more generally, or even matter when looked at from the perspective of the global climate. The purpose of this study was to explore the latter. Now that we have an idea of what matters, and what doesn't, the questions are better defined for the types of studies the reviewer had in mind.

Since both myself and Reviewer 2 commented on how Figure 5 was extremely hard to read when such similar symbols were used for both the 2 moment and 1 moment cases, it's odd that the authors did not update this figure for readability. Despite many different comments from all reviewers regarding the readability of the figures and the presentation of the results throughout the manuscript, I note that the only update to any of the figures in this revised version was a slight resizing of some of the panels in a few of the figures.

The layout of the figure and the symbols have been changed to improve readability, and these changes have been applied consistently across figures.

Even if readers can look up the details of the one and two moment schemes in their original paper, it seems extremely relevant to the current study that this information be readily available to the readers. I suggest that a table or figure be added that includes which process rates were included in each scheme, as requested by myself and multiple reviewers in the previous round of reviews.

In association with this study the 1-moment microphysics was completely rewritten. This identified a little over 200 parameters. The two moment scheme is yet more complex. While we appreciate the idea that somehow microphysics enumerates a well defined set of processes set by a few constants this is not the case in practice. Neither the categories are well defined and generalizable across models, nor are the processes and the parameters that regulate them few. Rather than persisting with this illusion by selecting a few of the constants and presenting them as if they describe the scheme, and because we don't believe a particular process rate is important for our arguments, we now outline this reasoning (§2.2) and recommend that the interested reader take the code itself (which is open source) as the ultimate documentation